# Double stranded RNA sensing is silenced during early embryonic development

Jeroen Witteveldt[1], Zicong Liu [2], Ana Ariza-Cosano [3,4], Christian Ramirez [5], Jessica L. Walters[1], Pilar G. Marchante [3,4], Lars Maas[2], Elias T. Friman[6], Alasdair Ivens[1], Toma Tebaldi[5,7], Sara R. Heras [3,4], Hendrik Marks [2] & Sara Macias [1] ✉

The type I interferon response is inactive during early mammalian development and becomes functional only after gastrulation. As a result, the totipotent and pluripotent embryonic stages remain susceptible to pathogens, including viruses. Here, we demonstrate that pluripotent mouse embryonic stem cells suppress the RIG-I-like receptor sensing pathway by silencing the expression of the double stranded RNA sensor MDA5. This silencing is necessary to avoid the recognition of double stranded RNAs of endogenous origin, which accumulate in mouse embryonic stem cells. Reintroducing MDA5 results in recognition of these endogenous double stranded RNAs and triggers the activation of the IFN response through IRF3. The production of interferon alters the differentiation ability of mouse embryonic stem cells, and affects the pluripotency gene expression programme, as shown by epigenetic, transcriptomic and proteomic analyses. Further, we show that zebrafish also repress MDA5 expression in early development and lack early-stage interferon activation, and that inducing double-stranded RNA-mediated signalling at this stage results in developmental defects. Altogether, we conclude that silencing the RIG-I-like receptor pathway during early development is important in preventing aberrant immune recognition of endogenous double stranded RNAs, safeguarding normal development.

Type I interferons (IFNs) orchestrate the innate immune response against viruses. IFNs are produced by infected cells, typically upon sensing viral nucleic acids. Viral cytoplasmic DNA is sensed by the cyclic GMP-AMP synthase, cGAS and viral RNAs are sensed by the retinoic acid-inducible gene I (RIG-I)-like receptor (RLR) family of proteins[1,2]. This family has three members, RIG-I, melanoma differentiation-associated protein 5 (MDA5) and laboratory of genetics and physiology (LGP2). RIG-I recognises uncapped short double-stranded RNAs (dsRNAs), while MDA5 recognises long dsRNA molecules. Although LGP2 binds a variety of RNA moieties, the absence of CARD domains prevents downstream signalling. LGP2 regulates the activity of the other two members in the family, preventing RIG-I activation, while promoting MDA5 activity[2]. Upon the sensing of viral nucleic acids, several families of transcription factors are activated,

[1]Institute of Immunology and Infection Research, Ashworth Laboratories, School of Biological Sciences, University of Edinburgh, Edinburgh, UK. [2]Department of Molecular Biology, Faculty of Science, Radboud University, Radboud Institute for Molecular Life Sciences (RIMLS), Nijmegen, The Netherlands. [3]GENYO, Centre for Genomics and Oncological Research: Pfizer/University of Granada/Andalusian Regional Government, Granada, Spain. [4]Department of Biochemistry and Molecular Biology II, Faculty of Pharmacy, University of Granada, Campus Universitario de Cartuja, Granada, Spain. [5]Laboratory of RNA and Disease Data Science, Department of Cellular, Computational and Integrative Biology (CIBIO), University of Trento, Trento, Italy. [6]MRC Human Genetics Unit, Institute of Genetics and Cancer, University of Edinburgh, Edinburgh, UK. [7]Section of Medical Oncology and Hematology, Department of Internal Medicine, Yale Comprehensive Cancer Center, Yale University School of Medicine, New Haven, CT, USA. ✉e-mail: smacias@ed.ac.uk

including the interferon regulatory factors 3 and 7 (IRF3/7), which drive type-I IFN expression, and the nuclear factor-kappa B (NF-kB), which drives the production of pro-inflammatory cytokines, such as TNF-α[3]. While IRF3 is expressed in most cell types, IRF7 has lower basal expression levels, that only increase upon IFN activation[4,5]. Besides type I IFNs, IRF3 also promotes the expression of other immune-related genes, suggesting that it has a broader range of targets than IRF7[6-8]. Once secreted, IFNs act in an autocrine and paracrine manner by binding to IFN receptors (IFNAR1/2) and inducing the expression of hundreds of interferon-stimulated genes (ISGs) via JAK-STAT signalling[9]. In addition to cytoplasmic sensors, the Toll-like receptors (TLRs), located on cell surfaces or endosomes, can also recognise viral nucleic acids and activate the type I IFN response[10].

Healthy mammalian cells typically do not accumulate significant amounts of endogenous dsRNAs, as they have evolved mechanisms to prevent their build up[11,12]. In contrast, certain pathological conditions, such as cancer or autoimmunity, are characterised by the activation of the IFN response upon sensing endogenous nucleic acids from nuclear or mitochondrial origins. This activation results from the malfunctioning of key RNA processing enzymes, as well as epigenetic regulators or nucleic acids sensors[13,14].

There is increasing evidence that transposable elements (TEs), due to their repetitive nature and their multiple insertions in either sense or antisense orientation, constitute a natural source of dsRNA in cells[15,16]. Most somatic cells silence transposon expression to prevent the potential deleterious consequences of novel insertions. Interestingly, embryonic stem cells (ESCs) and the early stages of development exhibit unique patterns of transposon activity. In contrast to somatic cells, early embryonic development is characterised by sequential waves of expression from specific TE families, such as LINE-1 in the mouse 2-cell stage, and MERVL in preimplantation mouse embryos[17]. These patterns are partly a result of the profound epigenetic reprogramming that embryonic cells undergo during the first divisions to acquire a totipotent and pluripotent identity, which typically suppresses TE activity. As an adaptation, embryonic cells exploit TEs, co-opting their promoters (e.g. long-terminal repeats, LTRs) and derived RNAs to fulfil essential functions during development[18-21].

ESCs and early developmental stages are also unique in aspects of immune defence. Both we and others have previously demonstrated that the IFN response is inactive in pluripotent ESCs and the early stages of mammalian development. When challenged with viruses or viral DNA/RNA mimics, both mouse and human ESCs are incapable of producing IFNs[22-25]. Similarly, mouse embryonic teratocarcinoma cells, which retain certain pluripotency traits, and human induced pluripotent stem cells also lack the capacity to produce IFNs[25,26]. The ability to synthesise IFNs is acquired after cells differentiate and lose their pluripotent state[22-25]. Supporting this, in vivo data show that in mice, the ability to produce IFNs is not acquired until after gastrulation, specifically at day 8 post fertilisation[27]. These observations lead to the hypothesis that the IFN response and pluripotency represent two incompatible cellular processes.

The critical question is why cells in early development, including ESCs, have sacrificed the type I IFN response if this renders them more susceptible to infections. While ESCs may use alternative antiviral pathways, we and others have shown that for specific viruses, ESCs are more permissive to infection than somatic cell types. This challenges the notion of a generalised intrinsic resistance of ESCs to viruses[24,28-32]. Within the inner cell mass, ESCs may be physically protected from viruses by the trophectoderm. In addition to the physical barrier, trophectoderm cells are immunologically competent and could stimulate the inner cell mass antiviral response by producing IFNs[33-35]. Although attenuated, ESCs can respond to exogenous IFNs, suggesting that maternal IFNs could also have protective properties during development[24,25,36,37]. For instance, constitutive expression of the type I

IFN, IFN-ε, by the maternal reproductive system has been shown to provide protection against viral infection[38].

To elucidate the mechanisms and rationale behind the absence of the type I IFN response in early development, we employed pluripotent mESCs and zebrafish as model systems. In both, we observed a silencing of the dsRNA-sensing pathway. Specifically, mESCs suppress the type I IFN response by silencing the expression of the dsRNA sensor MDA5. Unlike somatic cells, mESCs accumulate endogenous dsRNAs, that can induce the IFN response through MDA5 sensing. As a consequence, introducing MDA5 into mESCs activates the IFN response in the absence of infection. This inappropriate activation is highly detrimental, impairing ESCs ability to differentiate and maintain pluripotency, as shown by epigenetic, transcriptomic and proteomic analyses. The silencing of the dsRNA-sensing mechanisms appears to be a conserved evolutionary strategy amongst jawed vertebrates. Similarly, in zebrafish, MDA5 expression is low during the early stages of development, and activation of the IFN response results in obvious developmental defects.

We conclude that the gene expression program required for early development is inherently incompatible with an active IFN response. Expression of TEs is crucial for normal development but can lead to the formation of dsRNAs, potentially triggering the IFN response through MDA5 sensing. Our results indicate that IFNs disrupt normal development due to conflicts between IFN-associated transcription factors and the chromatin landscape in early development. Therefore, embryos must inactivate dsRNA sensing to suppress the IFN response, which, as a trade-off, increases their susceptibility to infections.

## Results

### The expression of the dsRNA sensor MDA5 is developmentally regulated

ESCs fail to make IFNs when challenged with dsRNA, but they acquire this ability after differentiation[22-24,39-42]. To identify which components of the IFN response may be responsible for the differential ability in producing IFNs, we compared the expression of genes involved in TLR signalling, RLR signalling and JAK/STAT signalling during stem cell differentiation. We used two different datasets, one of mouse ESCs differentiated to the three major embryonic lineages, ectoderm, mesoderm and endoderm[43], and a second, where mouse ESCs were differentiated to neuronal stem cells and neurons[44] (Fig. 1a–c and Supplementary Fig. 1A, B). Although varying between lineages, we observed a consistent increase in the levels of expression of *Ifih1*, the gene encoding for the dsRNA sensor MDA5, and *Irf7*, which encodes for the transcription factor IRF7. The expression of both genes was negligible in ESCs and increased after differentiation (Fig. 1a–c and Supplementary Fig. 1A, B). Both genes are involved in activating IFN production upon dsRNA stimulation, suggesting that the RLR pathway is silenced at multiple levels in ESCs. We confirmed these findings using in vitro differentiation assays of ESCs through induction of embryoid body formation followed by retinoic acid treatment[45]. After 20 days of differentiation, the expression of pluripotency factors *Nanog* and *Pou5f1* was lost, while the expression of differentiation markers *Sox17* and *Foxa2* increased as expected (Fig. 1d, e). We observed that *Ifih1* was the most differentially expressed gene involved in the RLR signalling pathway upon differentiation, displaying a significant increase (Fig. 1f). To evaluate potential mechanisms that drive silencing of *Ifih1* in ESCs, we first tested the role of DNA methylation. ESCs were grown in the presence of 5-azacytidine, a hypomethylation-inducing reagent, but no increase in *Ifih1* expression was found, while the known methylation-regulated gene *Dazl* was increased (Supplementary Fig. 1C). We next tested the contribution of polycomb-mediated silencing. RING1B, a component of the polycomb repressive complex 1 (PRC1), was found to bind the promotor region of *Ifih1* (Supplementary Fig. 1D). Both RING1B and homolog RING1A are

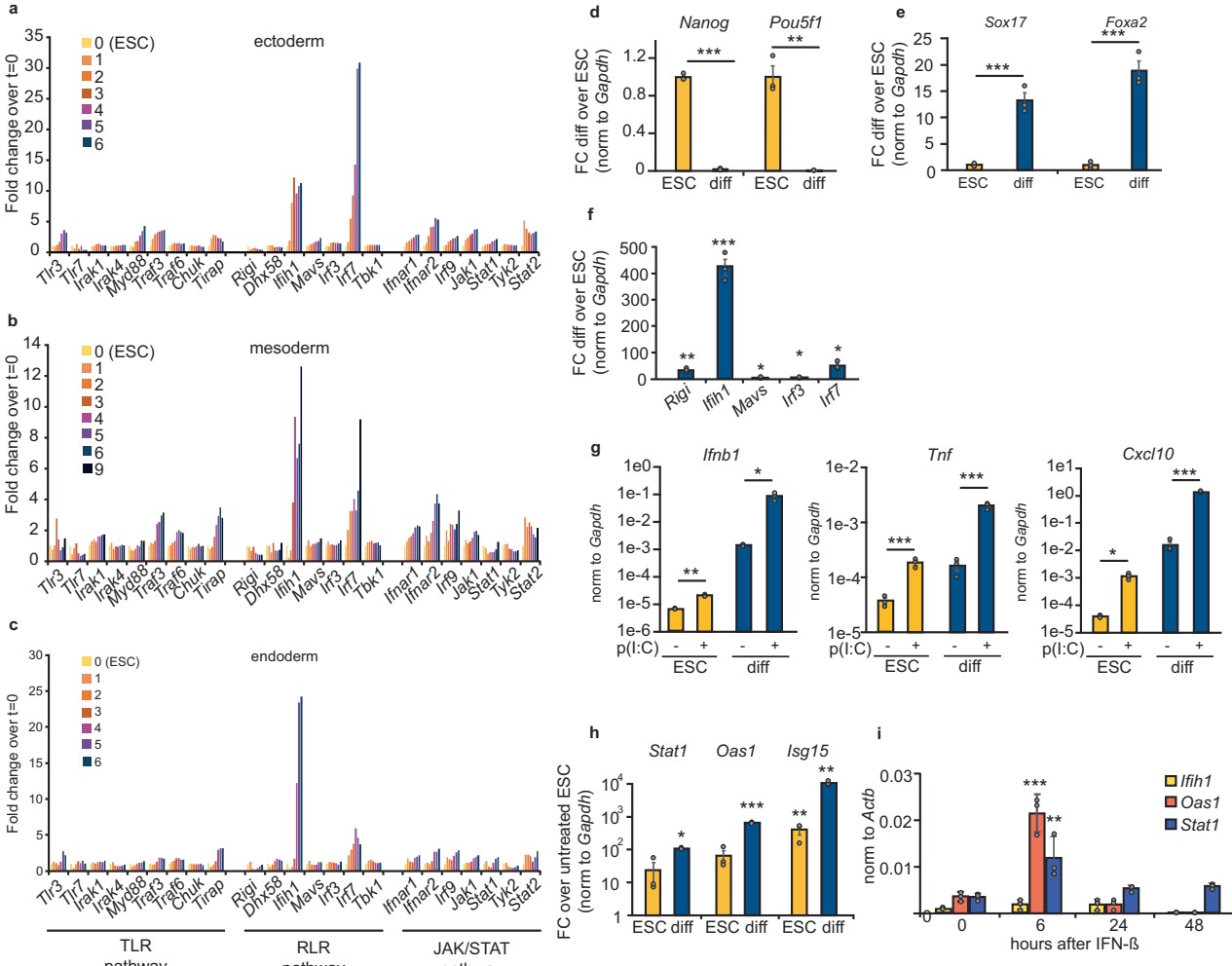

**Fig. 1 | Ifih1 expression is induced upon differentiation.** Expression of genes involved in TLR signalling, RLR signalling pathway and JAK/STAT signalling in ESCs differentiating to ectoderm (**a**), mesoderm (**b**) and endoderm (**c**), as obtained from RNA-seq. Samples were taken at successive days after starting differentiation (day 0 corresponds to ESCs, until day 6 or 9 of differentiation). Expression is calculated as normalised counts and relative to ESCs (day 0). RT-qPCR analyses for selected pluripotency genes (**d**), differentiation markers (*Nanog* $p = 0.000001$; *Pou5f1* $p = 0.001$; *Sox17* $p = 0.001$; *Foxa2* $p = 0.0007$) (**e**) and RLR signalling pathway genes (*Ifih1* $p = 0.0001$; *Rigi* $p = 0.006$; *Irf3* $p = 0.01$; *Irf7* $p = 0.05$; *Mavs* 0.04) (**f**) after embryoid body differentiation (Diff) of pluripotent ESCs (ESC). Data in **d**–**f** represent the average of three biological replicates ± the standard error of the mean (SEM). Single factor ANOVA was used to calculate significant differences amongst comparisons, followed by an *F*-test for variance and appropriate two-tailed *t*-test. **g** RT-qPCR analyses of *Ifnb1* (ESC $p = 0.006$; diff $p = 0.03$), *Tnf* (ESC $p = 0.002$; diff $p = 0.0004$) and *Cxcl10* (ESC $p = 0.02$; diff $p = 0.001$) upon dsRNA stimulation with poly(I:C) (p(I:C)) of ESCs and in vitro differentiated ESCs (as in **d**). Replicates and statistical tests as in (**d**). **h** Both ESCs and in vitro differentiated cells (as in **d**) were stimulated with exogenous IFN-β (100 pg/ml), ISGs induction was quantified by RT-qPCR. (*Stat1* diff $p = 0.006$; *Oas1* diff $p = 0.000004$; *Isg15* ESC $p = 0.03$ diff $p = 0.0001$). Replicates and statistical tests as in (**d**). **i** ESCs were stimulated with exogenous IFN-β (100 pg/ml), and ISG induction, including *Ifih1* ($t = 0$ vs $t = 6$; $p = 0.0004$), *Oas1* ($t = 0$ vs $t = 8$; $p = 0.01$) and *Stat1* was measured by RT-qPCR. Data represent the average of three biological replicates ± SD. One-way ANOVA was used to calculate significant differences amongst comparisons, followed by an F-test for variance and Tukey HSD. *$p$-val ≤ 0.05, **$p$-val ≤ 0.01, ***$p$-val ≤ 0.001.

functionally redundant and required for PRC1 function[46]. Despite binding, RING1B degron-depletion in *Ring1A*[−/−] ESCs did not result in a significant upregulation of *Ifih1* expression, suggesting it is not directly repressed by PRC1[47,48] (Supplementary Fig. 1E).

We confirmed that upon differentiation, and increased *Ifih1* expression, mESCs acquire the ability to respond to the dsRNA mimic, poly(I:C), as measured by RT-qPCR of IFNs (*Ifnb1*), *Tnf* as well as of the ISG *Cxcl10* (Fig. 1g). As sensing type I IFN is integral to the overall IFN response, we also compared the response of ESCs and differentiated cells to exogenous IFN-β. ESCs displayed an attenuated response to exogenous IFN-β, which increased upon differentiation, as shown by RT-qPCR analyses of a panel of ISGs (Fig. 1h). Despite some responsiveness to exogenous IFNs, *Ifih1* was not induced compared to other ISGs (Fig. 1i). All these together confirmed that ESCs acquire the IFN

response during differentiation, and this coincides with a significant upregulation of the dsRNA sensor MDA5 (*Ifih1*).

## Introduction of MDA5 in ESCs leads to developmental disruption

MDA5 is expressed at very low levels in ESCs, therefore, we decided to explore the consequences of expressing MDA5 in these cells by generating ESC lines constitutively expressing MDA5. Although MDA5 expression was driven by a promoter known to be stable in mESCs (EF1a), after several passages, cells silenced the exogenous copy of MDA5 in an irreversible manner. To address this issue, we generated doxycycline-inducible ESCs lines expressing a FLAG-tagged form of MDA5. Two clones were selected for further characterisation, and MDA5 expression after doxycycline treatment was confirmed both by

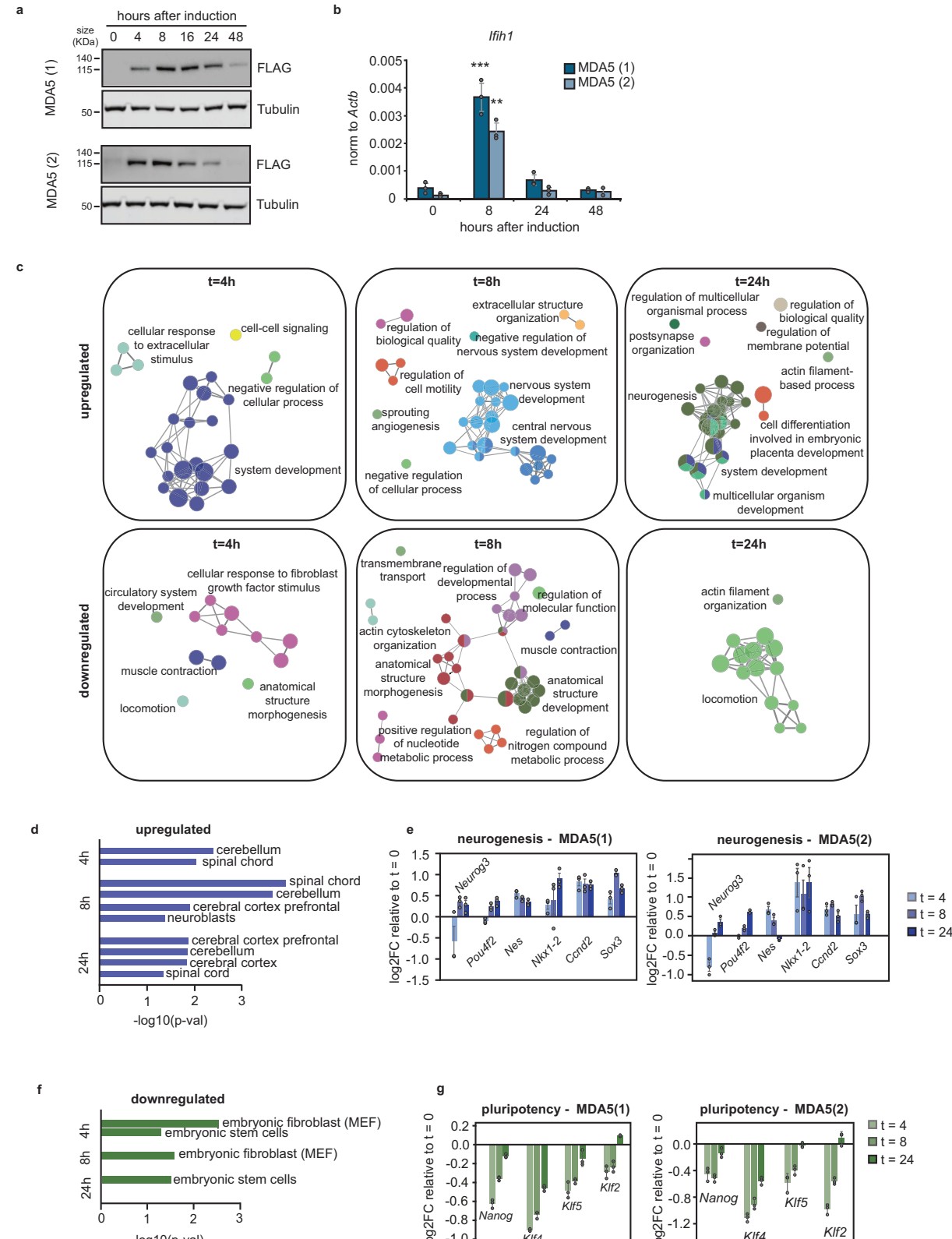

western-blot and RT-qPCR (Fig. 2a, b, respectively). Despite MDA5 expression reaching levels similar to those obtained after differentiation of cells (Supplementary Fig. 2A), we still observed silencing of MDA5 after 24 h, suggesting an adverse effect of long-term expression of MDA5. To further investigate the consequences MDA5 expression in ESCs, we performed a time course analysis by RNA-high-throughput sequencing (RNA-seq) after MDA5 expression (0, 4, 8, 24 h after

doxycycline induction) with two independent clonal cell lines compared to wild type ESCs (WT). Surprisingly, hundreds of genes were commonly differentially expressed in both MDA5-expressing cell lines compared to WT, with significant changes as early as 4 h after induction (abs log2FC > 0.4; $p$val ≤ 0.05) (Supplementary Data 1). Gene ontology (GO) analysis of the shared differentially expressed genes showed that both upregulated and downregulated genes were

**Fig. 2 | Ifih1 results in defects of the pluripotency and differentiation programme of ESCs. a** Two different ESC clones expressing a doxycycline-inducible N-terminal FLAG-tagged *Ifih1* gene were used. Upon doxycycline stimulation, *Ifih1* (MDA5) expression was confirmed by western blot (**a**) with anti-FLAG antibody. Tubulin serves as a loading control and expression was confirmed by RT-qPCR (**b**) of *Ifih1* (*t* = 0 vs *t* = 8; MDA5(1) *p* = 0.000001; MDA5(2) *p* = 0.000002). Data represent the average of three biological replicates ± SD. One-way ANOVA was used to calculate significant differences amongst comparisons, followed by an *F*-test for variance and Tukey HSD. *p*-val ≤ 0.05, **p*-val ≤ 0.01, ***p*-val ≤ 0.001. **c** Differentially upregulated and downregulated genes (abs log2FC > 0.4, *p*-val ≤ 0.05) common to the two *Ifih1*-inducible clones were used for gene ontology analyses (biological process). **d** Shared upregulated genes for both clones were analysed using the Mouse Gene Atlas for predicted cell types. *P*-value is computed using the Fisher exact test as proportion test assuming a binomial distribution and independence of the genes. **e** Log2FC expression changes ± SD of neurogenesis-related genes following MDA5 induction in clones MDA5(1) and MDA5(2) were determined using RNA-seq data from three independent biological replicates. **f** Shared downregulated genes for both clones were analysed using the Mouse Gene Atlas for predicted cell types. Statistics as in (**d**). **g** Log2FC expression changes ± SD of pluripotency-associated genes following MDA5 induction in clones MDA5(1) and MDA5(2) were determined using RNA-seq data from three independent biological replicates.

associated with development, morphogenesis, and differentiation, confirming the profound rewiring of the developmental programme when MDA5 is expressed (Fig. 2c, Supplementary Fig. 2B and Supplementary Data 2). More specifically, the upregulated genes were associated with terms involving nervous system development. This observation was confirmed using the Mouse Gene Atlas to predict cell types based on differential gene expression data, and by analysing the expression of specific neuronal markers by RNA-seq and RT-qPCR, suggesting that MDA5 induction in ESCs results in a 'neuronal-like' gene expression profile (Fig. 2d, e and Supplementary Fig. 2C). Conversely, downregulated genes were associated with genes typically expressed in ESCs and embryonic fibroblasts, suggesting that MDA5 induces the loss of the ESC features (Fig. 2f, g). We conclude that the expression of MDA5 causes a profound disruption of the gene expression programme of ESCs, leading to an expression profile associated with differentiating cells.

## MDA5 expression results in the loss of pluripotency factors and activation of immune pathways in ESCs

Considering the alterations in gene expression observed upon MDA5 induction, we hypothesise that these could be driven by epigenetic changes, including chromatin accessibility and/or differential activity of transcription factors. To test this possibility, we first assessed changes in the chromatin architecture upon MDA5 induction. We performed an Assay for Transposase-Accessible Chromatin coupled to high-throughput sequencing (ATAC-seq) and confirmed that MDA5 induces changes in chromatin accessibility, with 171 sites being more accessible, and a total of 382 losing accessibility after only 24 h of MDA5 induction (Fig. 3a, Supplementary Fig. 3A–C and Supplementary Data 3). Gene Ontology analysis revealed that regions with more accessibility were associated with nervous system function, similar to the RNA high-throughput sequencing analyses, but also deleterious pathways such as apoptosis. On the other hand, regions losing accessibility were associated with signalling pathways regulating the pluripotency of stem cells, including TGF-β, Apelin and p53 (Fig. 3b). Interestingly, regions changing accessibility were enriched for transcription factor binding sites of pluripotency factors, including Oct4, Nanog, and several Sox and Klf family members (Supplementary Fig. 3B). These findings were confirmed at the transcriptional level using the RNA-seq dataset. Using the loss-of-function (LOF) tool from the ShinyGO package[49], we found that the gene expression changes associated with MDA5 induction were predicted to be driven by the loss of transcription factors involved in pluripotency maintenance, such as Nanog, Oct4 (*Pou5f1*), and some members of the Polycomb repressive complexes 1 and 2 (PRC1 and PRC2), including Suz12, Eed and Rnf2 (Fig. 3c, for full list see Supplementary Fig. 3D). PRCs repress the expression of lineage-specific markers during pluripotency and silence the expression of pluripotency factors upon differentiation[50,51]. In agreement with this prediction, we confirmed that the expression of the pluripotency factors *Nanog*, *Klf4* and *Pou5f1* decreased upon MDA5 induction. In contrast, *Sox2* expression increased (Fig. 3d, e). *Sox2* has been shown to increase during neuroectodermal differentiation, which agrees with the neuronal-like gene expression phenotype that ESCs

acquire after MDA5 induction (Fig. 2d, e)[52,53]. Altogether, our combined analyses suggest that the function and expression of transcription factors essential for early development are affected by MDA5.

To provide further evidence of transcription factor dysregulation at the protein level and expand on the mechanism, we performed Chromatin-Enrichment for Proteomics followed by Mass Spectrometry (ChEP-MS). This approach provides a global overview of the chromatin composition and how these changes under different conditions[54,55]. The chromatin-associated proteome underwent substantial changes after 8 h of MDA5 induction, with comparable numbers of proteins showing increased or decreased presence (absLog2FC > 1, *p* < 0.05). Similar patterns were also observed after 24 and 48 h of MDA5 induction (Fig. 3f and Supplementary Data 4). KEGG pathway analyses revealed that enriched proteins in chromatin were associated with immune and pro-inflammatory signalling pathways after 24 h of MDA5 induction, while at 48 h, pathways involved in shutting down pro-inflammatory and IFN responses were activated[56,57] (Fig. 3g). On the other hand, factors involved in Notch, Hippo and p53 signalling, which are important for normal development, were depleted from the chromatin fraction after MDA5 induction (Fig. 3g). These results were reminiscent of the pathways obtained by ATAC-seq (Fig. 3b). We also exploited these datasets to understand the dynamics of key pluripotency maintenance factors. OCT4 and members of the Myc and Klf families, as well as PRC components, were less enriched in the chromatin fraction after inducing MDA5. In contrast, factors involved in de novo DNA methylation were chromatin-enriched upon MDA5 induction (Fig. 3h). To investigate if some of the changes observed in chromatin enrichment were driven by changes in gene expression, we compared a late ChEP-MS datapoint with an early RNA-seq timepoint. This comparison revealed a significant positive correlation (*p* = 0.003, *ρ* = 0.46) between RNA and protein levels, suggesting that the observed changes are not solely due to alterations in the subcellular localisation of chromatin factors, but also reflect shifts in the gene expression profile of these cells (Fig. 3i and Supplementary Fig. 3E, F). Interestingly, the group of shared genes with changes in the RNA and protein level was associated with innate immune signalling, including the ISG DDX58 (Fig. 3j). We predict that expression of MDA5 in ESCs results in innate immune activation, and a concomitant loss of pluripotency, accompanied by epigenetic and transcriptional reprogramming.

## Expression of MDA5 results in type I IFN activation upon recognition of endogenous dsRNAs

Both the high-throughput proteomic and RNA analyses suggested that MDA5 expression results in immune activation in ESCs in the absence of infection. To confirm this finding, we tested the expression levels of the type I IFN *Ifnb1*, *Tnf*, and the ISGs *Isg15* and *Ifitm1* in MDA5-induced cells using RT-qPCR. MDA5 induction resulted in the expression of type I IFNs and ISGs (Fig. 4a). On protein level, we observed production of IFN-β by ELISA in cells expressing inducible or constitutive forms of MDA5 (Fig. 4b). To test the importance of dsRNA recognition in activating the IFN response, we generated ESCs that express an MDA5 mutant form lacking the ability to recognise dsRNA. The C-terminal

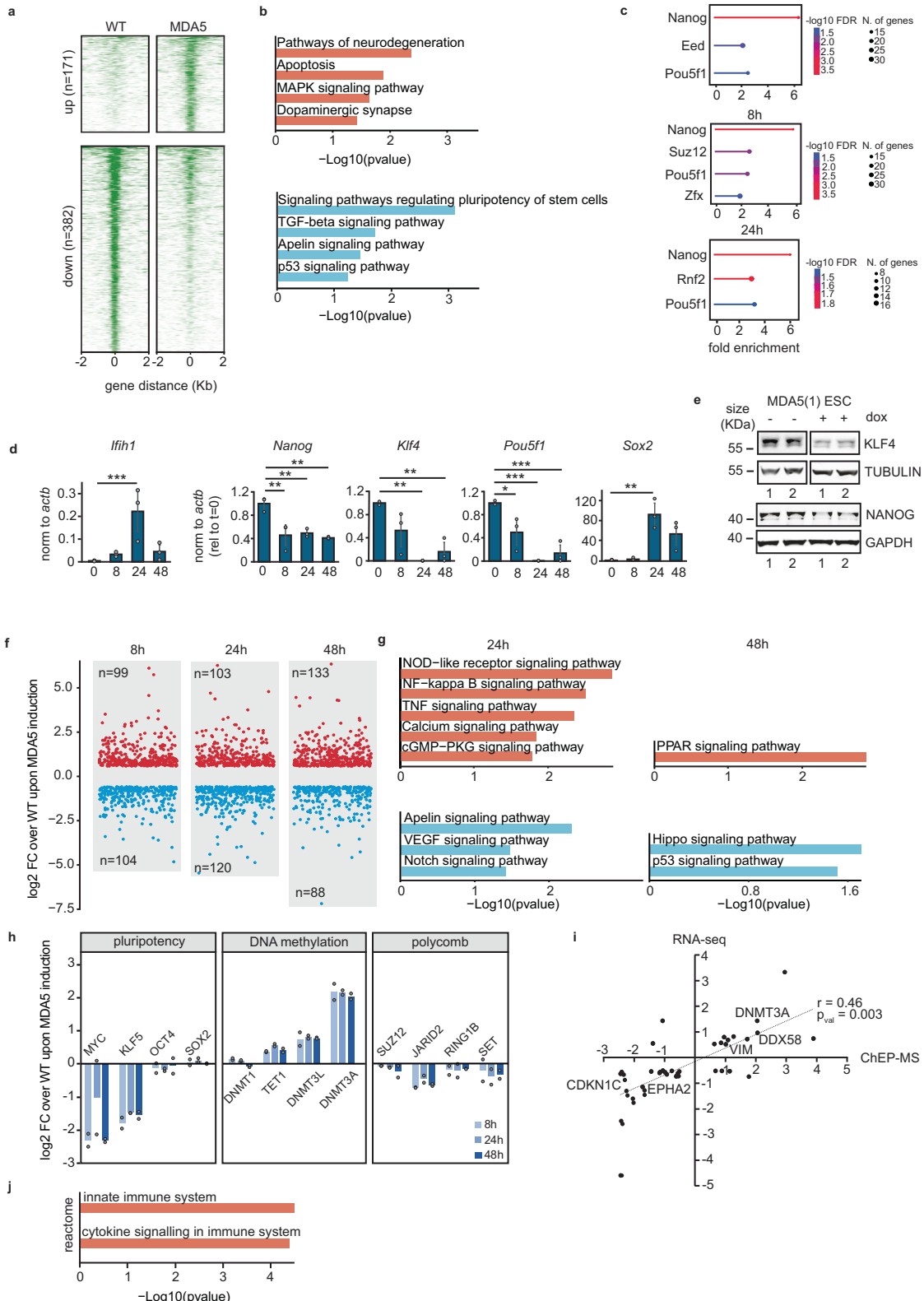

domain (CTD, residues 898–1020) of MDA5 is proposed to play a key role in the initial binding and oligomerisation on dsRNA[58,59]. Based on these findings an MDA5 variant lacking the CTD was generated (ΔCTD). Using in vitro pull-down assays, we confirmed that full-length MDA5 binds dsRNA (poly(I:C)), but not ssRNA (poly(C)), while the ΔCTD mutant failed to bind both dsRNA and ssRNA (Fig. 4c). We generated ESC lines expressing the inducible ΔCTD mutant and found that

expression of this truncated MDA5 form was not repressed as observed for full-length MDA5 (Fig. 4d). In addition, this mutant failed to activate the type I IFN response and consequent ISG expression in comparison with full-length MDA5 (Fig. 4b, e), suggesting that MDA5 requires binding to endogenous dsRNA and oligomerisation to initiate the IFN response. Based on these results, we tested whether ESCs accumulate endogenous dsRNA by performing flow cytometry with

**Fig. 3 | MDA5 expression leads to pluripotency transcription factor down-regulation and innate immune signalling. a** Differential accessible sites (ATAC-seq) for WT and MDA5 expressing cells after 24 h doxycycline treatment. Data represent the average of two biological replicates. **b** KEGG pathway analyses of genes associated with differential ATAC peaks, with the more accessible (up, top) and less accessible (down, bottom) regions upon MDA5 expression. Data is analysed using a Hypergeometric test and the *p*-value adjusted using the Benjamini-Hochberg method. **c** Differentially expressed genes (RNA-seq) at $t = 4$, 8 and 24 h after *Ifih1* induction were used to compute significant similarities with loss-of-function (LOF) datasets for transcription factors (TFs) in mESCs. **d** RT-qPCR analyses of pluripotency genes during *Ifih1* induction. Data represent the average of three biological replicates ± SD. One-way ANOVA was used to calculate significant differences amongst comparisons, followed by an F-test for variance and Tukey HSD. *p*-val ≤ 0.05, **p*-val ≤ 0.01, ***p*-val ≤ 0.001. Refer to Source Data file for exact *p*-

values. **e** Western blot analysis of KLF4 and NANOG protein levels in two independent biological replicates (1, 2) of uninduced MDA5(1) ESCs (-dox) and doxycycline treated for 14 h (+dox). Tubulin and GAPDH serve as loading controls. **f** ChEP-MS analyses of *Ifih1* induction with the number of significantly differentially enriched proteins for each timepoint indicated (two-sided Student's *t*-test, $P < 0.05$, FC > 2, 2 replicates). **g** KEGG pathway analyses for differentially enriched chromatin-associated proteins by ChEP-MS (two-sided Student's *t*-test, $P < 0.05$, FC > 2) at 24 h and 48 h. No significant terms were obtained at 8 h. **h** Comparison of the differential chromatin association of key regulators of pluripotency, DNA methylation and polycomb repressive complex upon MDA5 overexpression in mESCs ($n = 2$. Data are presented as the average ± s.e.m.). **i** Correlation between differential chromatin enrichment levels (ChEP-MS at 48 h) and RNA steady-state levels (at 8 h) upon MDA5 induction. Spearman rank correlation coefficient; $\rho = 0.46$, $p = 0.003$. **j** Reactome analysis of genes plotted in (**i**), statistics as in (**b**).

antibodies against dsRNA (J2). As a control for staining specificity, we pre-treated cells with RNAse III, a dsRNA-specific endonuclease, before probing with J2 antibodies. We confirmed that dsRNA accumulated in ESCs and this signal was RNAse III sensitive (Fig. 5a, b, and Supplementary Fig. 4A). On the other hand, the somatic IFN-competent microglial cell line, BV2, did not accumulate any RNAse III-sensitive dsRNA (Fig. 4a, b). To further confirm the presence of dsRNA in ESCs we performed immunofluorescence analyses in ESCs and in vitro differentiated cells. While differentiated cells did not display any dsRNA-positive cells, ESCs did, confirming that dsRNA is accumulated during pluripotency (Fig. 5c–e). We next aimed at identifying the transcripts present in the dsRNA population of ESCs using immunoprecipitation (IP) with J2 antibody followed by RNA high-throughput sequencing. We performed enrichment analyses by comparing the transcripts enriched in the dsRNA IPs ('enriched dsRNA') over input, versus all non-enriched transcripts ('non-enriched control'). This comparison revealed that dsRNAs were enriched in transcripts from protein-coding genes (Fig. 5f). Next, we calculated the minimum free energy (MFE) of the set of enriched transcripts and found a significant decrease compared to all non-enriched ones, confirming the presence of more structured conformations, possibly dsRNA (Fig. 5g). Both sets of transcripts displayed similar GC content, indicating that the lower MFE is not merely the consequence of a higher GC content (median %GC = 49.1 in non-enriched control, 48.9% in enriched dsRNAs) (Fig. 5h). Interestingly, the dsRNA-enriched transcripts were longer than the controls (2866 vs 1894 median nucleotides) (Fig. 5i). The main contributors for the lower MFE in dsRNA-enriched genes were the 3' UTR (untranslated regions) and coding sequence (CDS), and a significant increase in the proportion of intronic reads was found in the dsRNA-enriched transcripts (Fig. 5j, k). In agreement with the proposed roles for TEs in contributing to dsRNA formation in mammalian cells, and their enrichment in both 3'UTR and intronic regions[14], we found that the enriched dsRNAs contain significantly more TEs per gene, suggesting that TE-sequences contribute to dsRNA formation in ESCs (Fig. 5l). TE-enriched classes were both DNA and retrotransposons, from the SINE (B1, B2 and B4) and LTR (ERV-MaLR) class (Fig. 5m and Supplementary Fig. 5A).

These findings indicate that ESCs provide a permissive environment for dsRNA accumulation. However, this permissiveness is broken upon introduction of the dsRNA sensor MDA5, which activates the type I IFN response after dsRNA recognition.

### Who interferes with pluripotency: IFN production or IFN signalling?

Our results led us to hypothesise that IFN production is responsible for the dysregulation of pluripotency in ESCs. To test this possibility, critical genes for IFN production were depleted, and tested if MDA5 was no longer capable of perturbing pluripotency markers expression (Fig. 6a, b and Supplementary Fig. 6A). Only when IRF3 and MAVS were depleted, and no IFN response was activated, pluripotency genes

remained unchanged after MDA5 induction. On the other hand, depletion of IRF7 mimicked control cells, still showing decreased expression of pluripotency genes with MDA5 (Fig. 6c). This suggests that ultimately the activation of IRF3 and possibly the production of IFN is responsible for the loss in the expression of pluripotency genes.

To further confirm that pluripotency dysregulation is specific to MDA5 and IFN production, we measured the expression of pluripotency genes in MDA5-ΔCTD cells and MAVS overexpressing cells, which do not activate IFN production on its own. Neither of these resulted in perturbation of pluripotency genes nor activation of neuronal marker expression (Fig. 6d–f and Supplementary Fig. 2C). It therefore seems that the production of IFN is responsible for perturbing the pluripotency-associated gene expression programme. Finally, to test whether exogenous type I IFN stimulation could lead to a similar perturbation, ESCs were incubated with different amounts of IFN-β. This resulted in increased expression of ISGs such as *Oas1*, *Stat1* and *Isg15* by RT-qPCR, however, no significant changes in the expression of the pluripotency factors *Nanog*, *Klf4* or *Pou5f1* was observed (Fig. 6g). To further corroborate that IFN production rather than IFN signalling is responsible for the developmental dysregulation, we compared the expression of pluripotency genes in cells expressing MDA5 in the presence of Ruxolitinib. This compound is a well-studied JAK-STAT inhibitor that blocks intracellular IFN signalling[60]. As expected, no ISGs were induced in the presence of Ruxolitinib, despite a significant decrease in the expression of pluripotency genes (Fig. 6h). All these results together revealed that IRF3-mediated production of IFN, and not the sensing and signalling by exogenous IFN, is responsible for perturbing the pluripotency-associated gene expression programme.

### Suppression of the dsRNA sensing pathway is conserved in other vertebrates

Both mouse and human ESCs fail to produce IFNs when challenged with dsRNA, suggesting that the suppression of this pathway is conserved across mammals[32]. Jawed vertebrates, including fish and birds, also use the type I IFN response as the primary mechanism to defend against viruses. In zebrafish, the adaptive immune system is not well developed until 4 weeks post fertilisation, making it a useful model to study the innate antiviral response in the early stages of development[61–63]. Besides this, zebrafish are a powerful in vivo tool to assess the long-term effects of an active IFN response in early development on body plan formation. First, we confirmed that the expression of most RLR-signalling genes, including RIG-I, MDA5 and MAVS were also developmentally regulated in zebrafish (Fig. 7a)[64]. Next, we investigated whether this developmentally regulated expression correlated with the ability of zebrafish to respond to dsRNA. To this end, 1-cell embryos were injected with the dsRNA analogue poly(I:C), and IFN production was measured by in situ hybridisation with probes against the ISG *Trim25*. Only after 24 h did we observe an increase in *Trim25* staining (Fig. 7b) which was

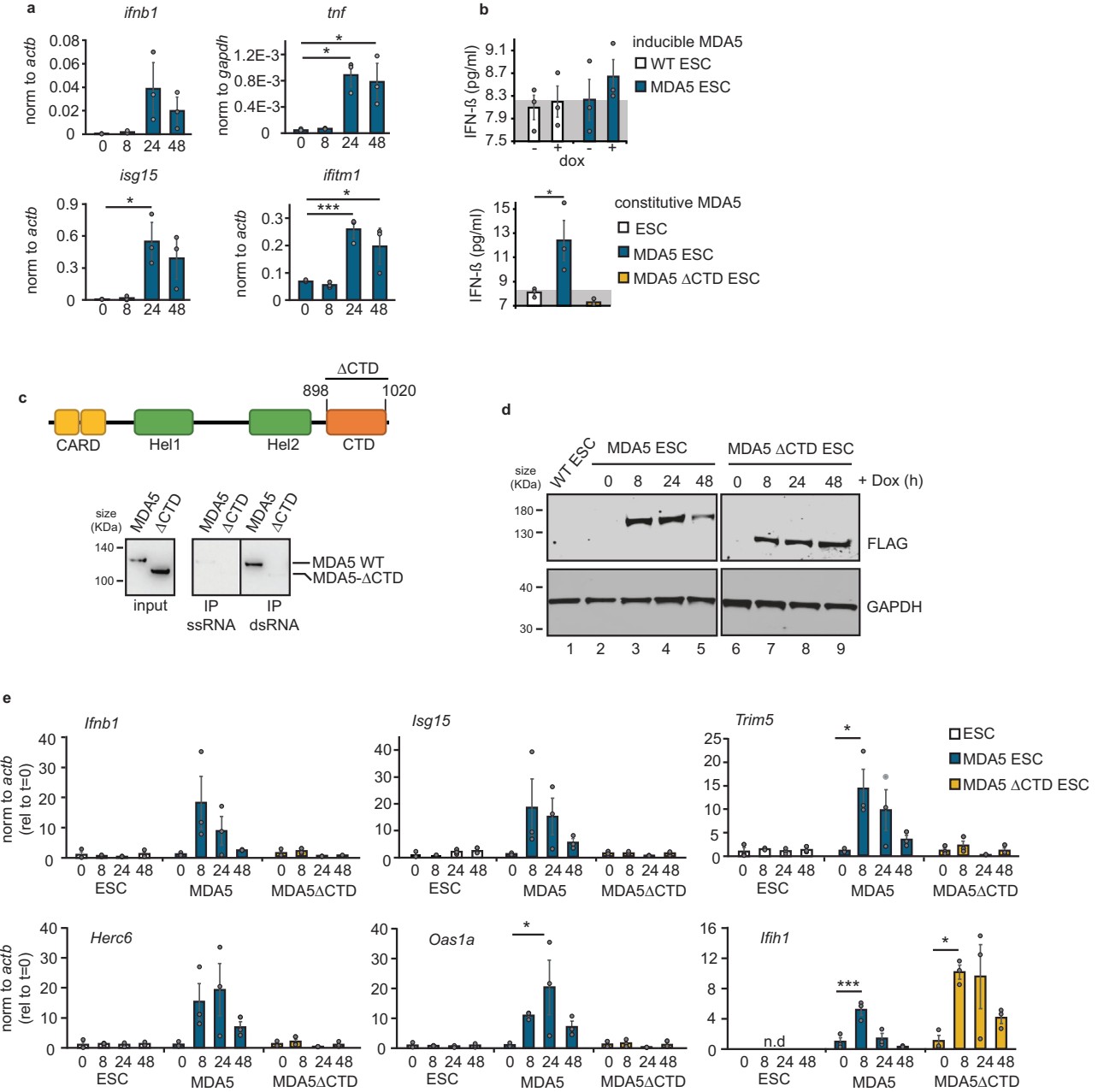

**Fig. 4 | MDA5 expression results in IFN activation through endogenous dsRNA recognition. a** RT-qPCR analyses of the type-I IFN *Ifnb1*, pro-inflammatory gene *Tnf* ($t = 0$ vs $t = 24$ $p = 0.007$; $t = 0$ vs $t = 48$ $p = 0.01$) and ISGs, *Isg15* ($t = 0$ vs $t = 24$ $p = 0.02$) and *Iftim1* ($t = 0$ vs $t = 24$ $p = 0.006$; $t = 0$ vs $t = 48$ $p = 0.05$) upon MDA5 induction time course. For *Ifih1* induction levels, see Fig. 3d. Data represent the average of three biological replicates ± SD. One-way ANOVA was used to calculate significant differences amongst comparisons, followed by an *F*-test for variance and Tukey HSD. *p-val ≤ 0.05, **p-val ≤ 0.01, ***p-val ≤ 0.001. **b** ELISA analysis of IFN-β production upon expression of inducible MDA5 ESCs (top) and constitutively expressed MDA5 (bottom). As negative controls, WT ESCs were treated with doxycycline (white) and cells constitutively overexpressing a mutant form of MDA5 (ΔCTD, yellow) were used. Cells constitutively expressing MDA5 produced significantly more IFN-β compared to control cells ($p = 0.04$). Grey area indicates detection limit of the assay. Number of replicates and statistical analysis as in (**a**).

**c** Schematic representation of MDA5 protein domains, CARD (caspase activation and recruitment domain), Hel1 (Helicase ATP-binding), Hel2 (Helicase C-terminal), C-terminal domain (CTD from residue 898 to 1020) (top). Western blot analyses of pull-downs with full-length (WT) and truncated (ΔCTD) MDA5 using ssRNA (poly(C)) and dsRNA (poly(I:C)) **d** Expression of MDA5 WT and ΔCTD in ESCs was confirmed by western blot analyses using anti-FLAG antibodies (top). GAPDH serves as a loading control (bottom). WT ESCs were included as a negative control (lane 1). All samples were run and probed on a single gel/blot (Supplementary Fig. 8.1 d) **e** RT-qPCR analyses of *Ifnb1* and ISGs induction after 8, 24, 48 h of doxycycline-inducible MDA5 WT (blue) and MDA5 ΔCTD (yellow) expression. WT ESCs were used as controls (white). For MDA5 expression levels, see (**d**). Number of replicates and statistical analysis as in (**a**), refer to Source Data file for exact p-values.

confirmed by analysing the expression of both *Trim25* and *Isg20* by RT-qPCR (Fig. 7c). DsRNA stimulation resulted in deleterious phenotypes in ~50% of the injected embryos. Some embryos exhibited impaired head development (class 1), while others showed other defects including issues with trunk development (class 2) (Fig. 7d, e). These results suggest that activating the response to dsRNA in early zebrafish development can also have negative consequences. We also investigated whether the dsRNA-induced phenotype was

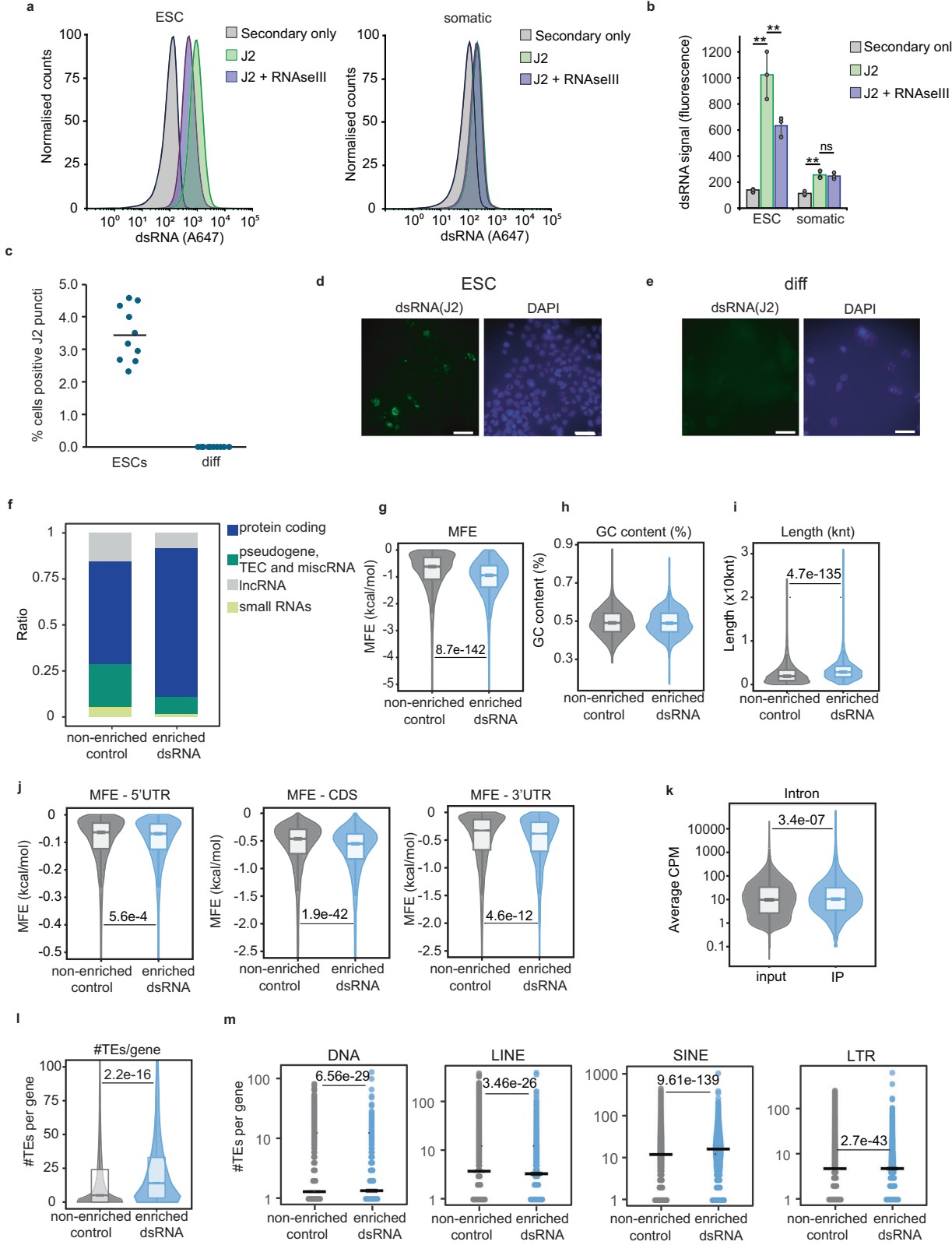

independent from IFN signalling and ISG production, as we previously observed in ESCs. To this end, dsRNA-injected zebrafish were treated with 25 µM of Ruxolitinib. The treatment resulted in decreased induction of *Trim25* (Fig. 7f). However, a similar number of fish displayed the dsRNA-induced phenotype with or without Ruxolitinib treatment (Fig. 7g). These results suggest that the dsRNA-induced developmental phenotype in zebrafish is independent of IFN

signalling, similarly to what we observed in ESCs (Fig. 6g, h). Finally, we also tested the consequences of IFN production in the differentiation ability of mESCs, as a model of in vitro development. IFN activation resulted in failure of mESCs to efficiently silence the expression of the major pluripotency factors during the first hours of differentiation, and premature expression of differentiating markers, such as *Hand1* (Supplementary Fig. 7A).

**Fig. 5 | ESCs accumulate endogenous dsRNA derived from transposable elements. a** Flow cytometry analyses using dsRNA specific antibodies (J2), in ESCs (left panel), and differentiated somatic BV2 cells (right panel). J2 + RNAse III, and secondary only samples serve as controls for antibody specificity and background, respectively. **b** Quantification of average dsRNA signal (n = 3, ±SD). ESCs express significant amounts of dsRNA (p = 0.004), which is reduced in the presence of RNAseIII (p = 0.0002). The lower dsRNA signal in somatic cells is not significantly reduced in the presence of RNAseIII. Data represent the average of three biological replicates ± SD. One-way ANOVA was used to calculate significant differences amongst comparisons, followed by an F-test for variance and Tukey HSD. \*p-val ≤ 0.05, \*\*p-val ≤ 0.01, \*\*\*p-val ≤ 0.001. **c** Quantification of ESCs and in vitro differentiated cells containing positive signal for J2 antibody (punctate appearance). Ten different microscopy images were quantified, data are represented as percentage of positive cells. Statistical analysis performed as in (**b**), p-value =

0.00000000001. Representative images for J2-based immunofluorescence of ESCs (**d**) and in vitro differentiated cells using retinoic acid (**e**). DAPI serves as nuclear stain. **f** Biotype distribution of transcripts enriched in dsRNA immunoprecipitations versus input compared to non-enriched controls. **g** Minimum Free Energy (mfe, kcal/mol) **h** % GC-content **i** distribution of transcript length (knt) **j** MFE from 5' untranslated regions (5' UTR), coding sequences (CDS) and 3'UTRs, of dsRNA-enriched vs non-enriched controls. **k** Average counts per million (cpm) mapping to introns in input vs dsRNA IP. **l** Distribution of the number of annotated TEs per transcript in dsRNA-enriched vs non-enriched controls. Box plots in **g**–**l** show First quartile ($Q_1$), Median ($Q_2$) and Third quartile ($Q_3$), with whiskers from Minimum ($Q_0$) to Maximum ($Q_4$). **m** From left to right, distribution of the number of DNA transposons and LINE, SINE and LTR retrotransposons in dsRNA-enriched vs non-enriched controls. Results and statistical analyses presented in **f**–**m** were derived from three independent biological replicates.

All these together suggest that the ability to synthesise IFNs in zebrafish is developmentally regulated and, like mice, activation of the dsRNA-mediated responses during early embryonic stages can have severe developmental consequences. We hypothesise that silencing of the dsRNA-sensing pathway is conserved across jawed vertebrates.

## Discussion

The importance of silencing the IFN response during early development is illustrated by the observation that the RLR pathway, the primary activator of IFN production, is suppressed at different levels and through multiple mechanisms. Our findings show that both the dsRNA sensor MDA5 (*Ifih1*) and the transcription factor *Irf7* are transcriptionally regulated during development, and increase their expression upon differentiation. Previously, overexpression of a catalytically active mutant form of IRF7 was also shown to result in IFN expression in ESCs, similar to MDA5[25]. Similar results have been recently obtained with IRF3 overexpression[65]. However, it is still unclear what the mechanisms are that lead to silencing of *Ifih1* and *Irf7* in ESCs. Our results show that neither DNA methylation nor PRC1 are required to silence *Ifih1* in ESCs[66]. It is possible that differentiation leads to induction of transcription factors which activate the expression of *Ifih1* and *Irf7*.

Transcriptional control is not the only mechanism silencing the IFN response in ESCs. We previously demonstrated that MAVS is post-transcriptionally silenced by miRNAs in mESCs. Removal of the MAVS-targeting miRNAs, or overexpression of MAVS restored the ability of mESCs to respond to dsRNA and increased their ability to defend from viruses[24]. This redundancy in suppression of the IFN response by targeting different components of the pathway suggests that its downregulation is absolutely critical during early development.

The suppression of the dsRNA-sensing pathway seems to be conserved across a range of species. Re-analyses of RNA high-throughput sequencing datasets demonstrate that other vertebrates also developmentally regulate the RLR pathway, including chickens (*Gallus gallus*), marmosets (*Callithrix jacchus*), macaques (*Macaca mulatta)*, frogs (*Xenopus tropicalis*) and zebrafish (*Danio rerio*)[64,67–70]. In line with the dysregulation observed in mESCs, IFN induction in early development in zebrafish leads to developmental defects. Others have also observed that zebrafish embryos can produce IFN-φ1 upon dsRNA stimulation at 12 h post fertilisation which is accompanied by a decrease in survival[71]. These results together with our time course analysis suggest that the IFN response in zebrafish is enabled between 6 and 12 h post fertilisation, coinciding with increasing levels of MDA5 and once the maternal-to-zygotic transition and gastrulation have occurred. A more detailed time course analyses will be required to find when exactly zebrafish gain an active IFN response.

Our results suggest that it is important to silence MDA5 to prevent endogenous dsRNA recognition and activation of the IFN response. The accumulation of dsRNA in ESCs was an unexpected finding, as

normally, the presence of dsRNA is associated with viral infections. ESC-dsRNA derived from protein-coding genes with strong secondary structures and enriched in TE-sequences. Previous studies have implicated TE-derived sequences as substrates for viral nucleic acid sensors, including MDA5[14,72,73]. Interestingly, the overexpression of MDA5 in other cell types does not always result in spontaneous activation of the IFN response, and it seems to be a cell-line dependent response, suggesting that cells accumulate different levels of unedited dsRNA[74–79]. Future efforts will aim at understanding the function of these dsRNAs in ESC biology and development.

As a result of dsRNA recognition, MDA5 triggers the IFN response and a profound rewiring of the developmental programme of ESCs. The temporal scale of these changes suggests that most are driven by alterations in transcription factor activity, localisation and/or post-translational modifications rather than de novo protein synthesis. In agreement, ATAC-seq and ChEP-MS confirmed that MDA5 induction results in rapid changes in chromatin accessibility and chromatin-enrichment of transcription factors associated with pluripotency, differentiation, and immune responses. Interestingly, we also observed changes in the chromatin-enrichment of epigenetic factors, including DNA methyltransferases (DNMT) and polycomb-associated proteins. It is possible that some of the gene expression changes observed after MDA5 induction are mediated by alterations of polycomb function. For instance, MDA5 induction mimics some of the molecular defects observed in polycomb-deficient mESCs, where expression of differentiation markers occurs during pluripotency[80]. MDA5-overexpressing ESCs also resemble PRC2-deficient mESCs (*Suz12*[−/−]), as they fail to efficiently repress pluripotency markers during differentiation[81]. A more detailed study of which epigenetic and transcription factors are altered by MDA5 will provide further mechanistic insights into the role of this sensor in pluripotency and differentiation interference.

We also explored which specific factors and steps of the IFN response are responsible for altering the biology of ESCs. Our results suggest that it is not MDA5 itself who is interfering with pluripotency, but rather IRF3 activation and production or transcription of IFN itself. Similarly, overexpression of IRF7 or IRF3 also results in dysregulation of the ESC's gene expression profile[25,65]. In agreement, we only observed alterations in the pluripotency gene expression programme when we forced cells to produce IFNs, but not upon sensing of exogenous IFN. This is possibly due to the necessity of embryos to still be able to respond to maternal IFNs during early development[82]. Previous studies have also hinted towards interactions between pluripotency factors and the IFN response. For instance, the ability of IRF7 to induce the IFN response in ESCs can be reversed by overexpressing the pluripotency factors Oct4, Sox2 and Klf4[25]. It is unknown if this observation is only specific to IRF7, and the exact mechanism by which these TFs can block IRF7 activity. Interestingly, we observed down-regulation of Nanog, Oct4 and Klf4 levels upon MDA5 induction, suggesting that changes in the expression of these factors could also

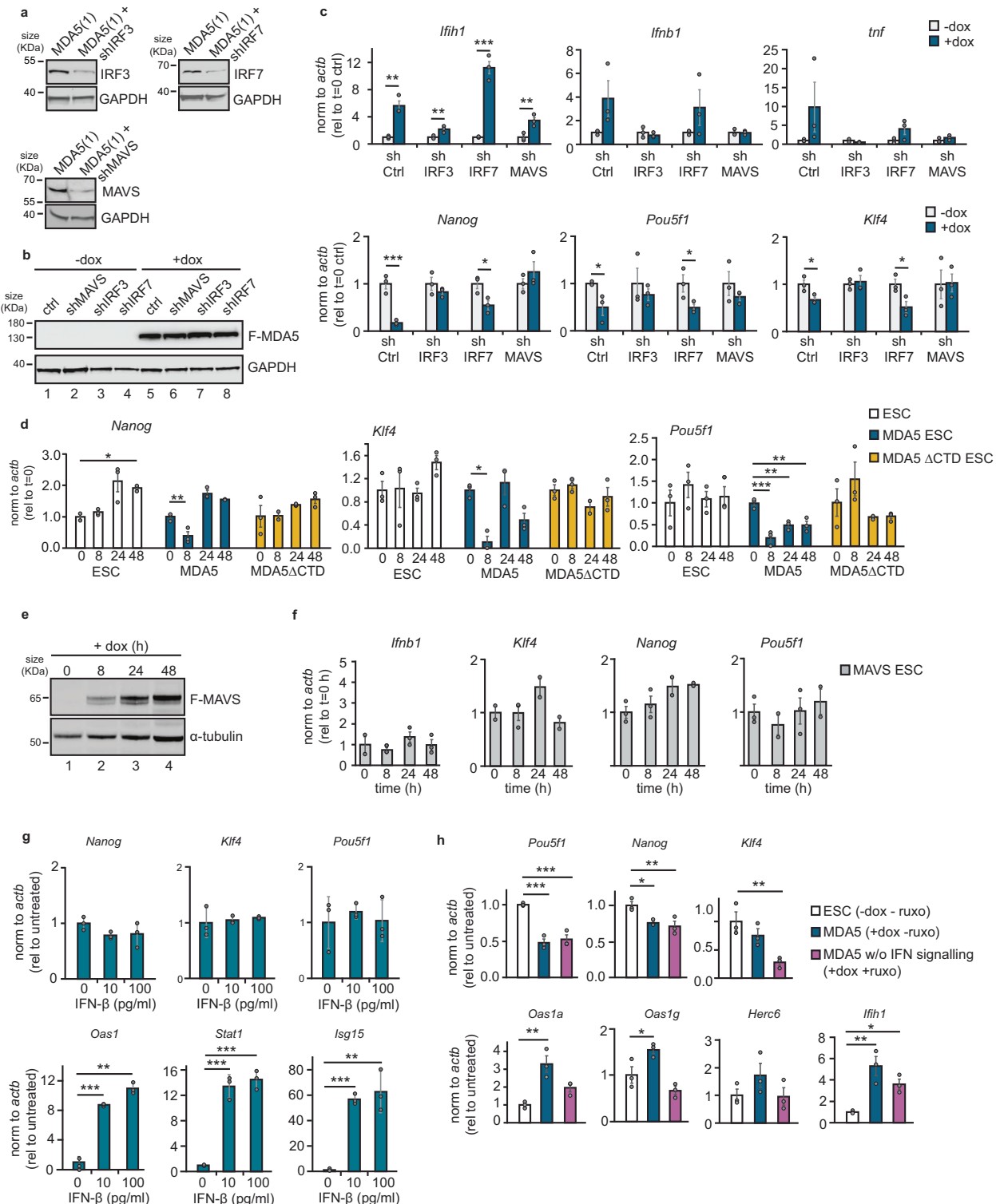

be responsible for the dysregulation of the pluripotency gene expression programme.

Importantly, the production of IFN is not only toxic during early development. A strong IFN response during later stages can also lead to developmental abnormalities. Congenital infections, where maternal infections are vertically transmitted to the foetus, are a major cause of neurodevelopmental complications[83]. Most infections occur post-implantation, affecting IFN-competent cells after evading maternal-placental defence systems. Similarly, interferonopathies—Mendelian diseases characterised by excessive IFN secretion—often lead to

neurological developmental issues despite IFN activation occurring later in development[13,84]. Given these patterns, it is reasonable to speculate that they share a common mechanistic basis with our findings in mESCs and the developmental issues of zebrafish.

To conclude, we hypothesise that the gene expression program required for early development is incompatible with an active dsRNA-based antiviral response. Production of IFNs is deleterious for development and results in a global disruption of normal pluripotency and development, likely due to the incompatibility between the active transcription factors and available chromatin landscape. To avert this

**Fig. 6 | IRF3 activation and IFN production is responsible for pluripotency perturbation. a** Western blot showing stable knockdown of IRF3, IRF7 and MAVS by shRNAs in MDA5(1) overexpressing ESCs. GAPDH serves as a loading control. **b** Western blot analysis of the same shRNA-expressing cells with/without doxycycline. GAPDH serves as a loading control. **c** RT-qPCR analyses for the MDA5 gene *ifih1*, type-I interferon *Ifnb1*, pro-inflammatory gene *Tnf* and pluripotency markers *Nanog*, *Pou5f1* and *Klf4* in the shRNA-expressing lines from (**a**) plus shCtrl, a non-targeting shRNA control. Refer to Source Data file for exact *p*-values. Data are the average of three biological replicates ± SEM, Single factor ANOVA was used to calculate significant differences amongst comparisons, followed by an *F*-test for variance and appropriate two-tailed *t*-test *$p$-val ≤ 0.05, **$p$-val ≤ 0.01, ***$p$-val ≤ 0.001. **d** RT-qPCR analyses of pluripotency factors *Nanog* (v6.5 $t = 0$ vs $t = 48$ $p = 0.03$); MDA5 $t = 0$ vs $t = 8$ $p = 0.007$), *Pou5f1* (MDA5 $t = 0$ vs $t = 8$ $p = 0.0004$; $t = 0$ vs $t = 24$ $p = 0.008$; $t = 0$ vs $t = 48$ $p = 0.008$) and *Klf4* (MDA5 $t = 0$ vs $t = 0$ $p = 0.006$) upon expression of doxycycline-inducible WT-MDA5 and mutant (ΔCTD) MDA5 in

mESCs. WT ESCs are included as a negative control. Data represent the average of three biological replicates ± SEM. One-way ANOVA was used to calculate significant differences amongst comparisons, followed by an *F*-test for variance and Tukey HSD. *$p$-val ≤ 0.05, **$p$-val ≤ 0.01, ***$p$-val ≤ 0.001. **e** Western blot of time course of ESCs expressing a doxycycline-inducible form of MAVS tagged with FLAG using anti-FLAG. Tubulin serves as a loading control. **f** RT-qPCR analysis of *Ifnb1* and pluripotency genes expression after MAVS induction in ESCs. Replicates and statistical tests as in (**d**). **g** RT-qPCR analyses of ISGs and pluripotency genes upon exogenous IFN-β stimulation of v6.5 cells using two different concentrations. Replicates and statistical tests as in (**d**), refer to Source Data file for exact *p*-values. **h** RT-qPCR analyses of pluripotency factors (*Pou5f1*, *Nanog*, *Klf4*) and ISGs (*Oas1a*, *Oas1g*, *Herc6*) in WT ESCs compared to cells overexpressing MDA5 (+dox -ruxo), and cells overexpressing MDA5 in the presence of 5 μM Ruxolitinib (+dox +ruxo). Replicates and statistical tests as in (**d**), refer to Source Data file for exact *p*-values.

---

disruption, cells have suppressed the IFN response at multiple levels, with the unavoidable consequence of making them more susceptible to viral infections.

## Methods

### Ethics statement

All experiments performed with zebrafish at the University of Granada comply with national and European Community regulations for the use of animals in experimentation and were approved by the ethical committees of the University of Granada and the Junta de Andalucía.

### Mouse and zebrafish datasets and analysis

Sequencing data from mouse ESC differentiation studies were obtained from the Geo Expression Omnibus data repository[44], GSE127741) and ArrayExpress data collection[43], E-MTAB-4904). FastQ files were extracted from the repositories and quality assessed using FASTQC (v0.11.8). Cutadapt (v3.4) was used to trim the sequences using the command [cutadapt -a AGATCGGAAGAG -A AGATCGGAA-GAG -j 0 -m 50 -o tr.$fq1 -p tr.$fq2 $fq1 $fq2 > tr.$sample.log]. Sequences were not collapsed within each sample, and duplicates were retained. Alignment of reads (--very-sensitive -p 24 --no-mixed --no-discordant --no-unal -x $db −1 tr.$fq1 −2 tr.$fq2 2» $bt2log) to the REL104 reference set was performed using hisat2 (v 2.2.1). Alignment rates were typically 90% of input reads per sample. Alignments were stored in sorted, indexed BAM files. Raw tag counts per gene in each sample were extracted from the BAM files using the gtf file in combination with featureCounts function of Rsubread (featureCounts(files = bamfiles, annot.ext = 'Mus_musculus.104.gtf.gz', isGTFAnnotationFile = TRUE, countMultiMappingReads = TRUE, allowMultiOverlap = TRUE, isPairedEnd = TRUE, nthreads = 24, minOverlap = 10)), and were normalised as part of the DESeq2-based groupwise comparisons: no conversions (e.g. reads per kb of gene, reads per million, etc.) were performed. The zebrafish dataset was obtained from the Geo Expression Omnibus data repository[64], GSE106430). Raw reads were aligned to the complete assembly of zebrafish genome GRCz11 and quantified according to NCBI RefSeq *D. rerio* Annotation Release 106 (June 2, 2017) using STAR v2.7.11b[85] and quantified according to NCBI RefSeq *D. rerio* Annotation Release 106 (June 2, 2017) using featureCounts function of Subread v2.0.8.

### Cells and differentiation protocols

The mouse ESC line v6.5 was obtained from ThermoFisher (MES1402) and cultured in Dulbecco's modified Eagle Medium (DMEM, Thermo-Fisher) supplemented with 15% heat-inactivated foetal calf serum (ThermoFisher), 1X Minimal essential amino acids (ThermoFisher), 2 mM L-glutamine, $10^3$ μ/ml of LIF (Stemcell Technologies) and 50 μM 2-mercaptoethanol (ThermoFisher). ESCs were grown on plates coated with 0.1% gelatine, detached using 0.05% Trypsin (ThermoFisher). The mouse microglial BV-2 cells were cultured in Dulbecco's modified

Eagle Medium (DMEM, ThermoFisher) supplemented with 10% heat-inactivated foetal calf serum (ThermoFisher) and 2 mM L-glutamine. Cells were incubated at 5% $CO_2$ at 37 °C.

Mouse ESCs were differentiated using hanging droplets and retinoic acid to induce embryoid body formation and differentiation as described before[45]. In brief, ESCs were washed and resuspended in culture media without LIF. Individual drops of 20 μl cell suspension are pipetted on non-culture treated Petri dish lids and cultured upside down for 48 h. During that time embryoid bodies (EBs) will form and are collected into non-culture treated plates for another 48 h before adding retinoic acid to the media (250 nM). After 7 days of further incubation, the EBs are moved to gelatine-coated cell culture treated plates where they will attach and are ready for subsequent experiments.

Differentiated cells were collected and RNA was extracted using Tri-reagent (Sigma-Aldrich) and processed for real time PCR as described below.

For all doxycycline (dox) treatments, 250 ng/ml concentration was used. For poly(I:C) treatment, differentiated cells were collected and seeded in 24 well plates, transfected with 2.5 μg poly(I:C) using lipofectamine 2000 and incubated for 6 h before collecting the cells in Tri-reagent. For exogenous IFN-β treatment, differentiated cells were collected and seeded in 24 well plates and incubated with 10 or 100 pg/ml of mouse IFN-β (Bio-techne, 8234-MB-010) for 4 h before collecting the cells in Tri-reagent. Ruxolitinib (Invivogen, tlrl-rux-3) treatment was performed at 5 μM concentration. To assess the impact of the IFN response during differentiation a modified differentiation protocol was used. V6.5 and MDA5-overexpressing cells were pre-treated with doxycycline for 8 h before a single-cell suspension of $5 × 10^5$ cells/ml was prepared in medium without LIF and in the presence or absence of doxycycline. Twenty μl drops were pipetted on the inside of the lid of a 10 cm petri dish and incubated upside-down at 37 °C, 5% $CO_2$ for 24 h. The embryoid bodies were consequently washed from the lids, pelleted by centrifugation and RNA extracted using Tri-reagent.

### RNA extraction and real time PCR

Total RNA was extracted using Tri-reagent (MilliporeSigma), and cDNA was synthesised using M-MLV (Promega) or the High-Capacity cDNA Reverse Transcription Kit (Applied Biosystems) in accordance with the manufacturer's instructions. qPCR reactions were performed using GoTaq qPCR mastermix (Promega) or qPCRBIO sygreen (PCR Biosystems) using previously published primers (Supplementary Data 5) on a QuantStudio 5 (ThermoFisher) or Azure Cielo 6 (Azure biosystems). Data was analysed using Quantstudio Design & Analysis software or Azure Cielo Manager software. Differences were analysed by single factor ANOVA to calculate significant differences amongst comparisons, followed by an F-test for variance and appropriate two tailed *t*-test *$p$-val ≤ 0.05, **$p$-val ≤ 0.01, ***$p$-val ≤ 0.001.

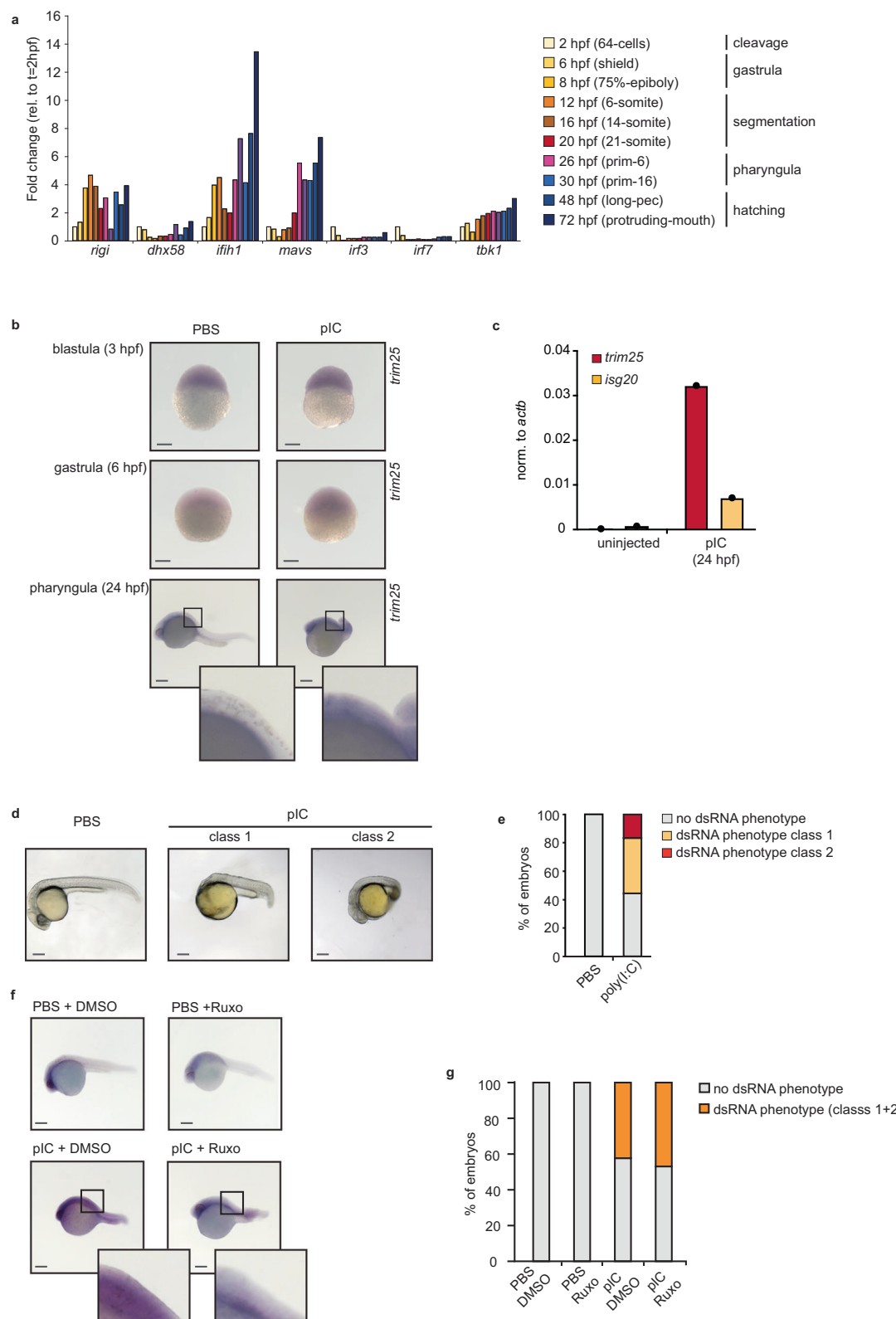

## Western blot assay

Whole-cell extracts were collected in RIPA buffer containing protease inhibitors followed by sonication and centrifugation for clarification of extracts. Extracts were quantified using the BCA assay, mixed with LDS sample buffer (Invitrogen) and Sample reducing agent (Invitrogen) and run on 4–12% Bis-Tris precast gels (ThermoFisher). Proteins were transferred to nitrocellulose membrane using semi-dry (iBlot2, ThermoFisher) or wet transfer (mini blot module, ThermoFisher) according to the manufacturer's recommendations. Membranes were blocked for 1 h at room temperature in PBS-T (0.1% Tween-20) and 5% milk powder before overnight incubation at 4 °C with primary antibody. Antibodies used were anti-mouse HRP (7076, CST), anti-Rabbit HRP (7074, CST), anti-mouse 680RD (925-68070, LICORbio), anti-mouse 800CW (926-32210, LICORbio), anti-rabbit 680RD (925-68071,

**Fig. 7 | IFN response is silenced in zebrafish early development. a** Relative expression of RLR-signalling pathway genes across zebrafish development. Each timepoint is expressed relative to the expression at the earliest timepoint (2 h post fertilization (hpf)). **b** Representative pictures of in situ hybridisation against the ISG *trim25* to monitor IFN activation 3, 6 and 24 h post-injection of dsRNA (polyI:C (pIC)) at the zygotic stage (0 hpf). PBS injection serves as a negative control. Three biological replicates using 200 embryos for each treatment were split in three timepoints and used for in situ hybridization. Scale bar is 200 μm. **c** RT-qPCR analyses of ISG expression at 24hpf after dsRNA stimulation with poly(I:C) using >50 embryos for each sample. **d** Representative pictures of phase-contrast microscopy of dsRNA-injected embryos (pIC, right), versus PBS-injected controls (left), 24 h after injection using the same embryos from (**b**). Scale bar is 200 μm.

**e** Developmental defects were quantified as 'dsRNA phenotype class 1' (failure to develop head), and 'dsRNA phenotype class 2' (failure to develop the trunk) and represented as the proportion of embryos with defects vs non-defective. **f** In situ hybridisation against the ISG *trim25* to monitor IFN activation 24 h post-injection of dsRNA (polyI:C) in the presence and absence of the JAK-STAT inhibitor, Ruxolitinib (Ruxo, 25 μM). PBS and DMSO injections serve as a negative control (vehicle only). Two-hundred embryos for each treatment were injected with PBS 1 h before the addition of DMSO or Ruxolitinib and fixed at 24hpf. The experiment was repeated twice. Scale bar is 200 μm. **g** Quantification of the dsRNA-induced phenotype upon poly(I:C) (pIC) injection in the presence and absence of Ruxolitinib (Ruxo). Data are represented as percentage of embryos with dsRNA phenotype.

LICORbio), anti-rabbit 800CW (926-32211, LICORbio) anti-FLAG (M2, Merck), anti-tubulin (CP06, Calbiochem), anti-GAPDH (CB1001, Merck), anti-Nanog (ebioMLC-51, Invitrogen), anti-KLF4 (4038, CST), anti-IRF3 (12A4A35, BioLegend), anti-IRF7 (MA5-52511, Invitrogen) and anti-MAVS (sc365334, Santa Cruz). Proteins bands were visualised using ECL (Pierce) on a Bio-Rad ChemiDoc imaging system or Li-cor imaging system (LICORbio), depending on the secondary antibody used. Protein bands were quantified using ImageJ (v1.53q) software and expression levels calculated normalised to tubulin.

## J2 flow cytometry and Immunofluorescence

Cells were dissociated using 0.05% Trypsin, washed in PBS and resuspended in FACS buffer (PBS with 1% FBS). Cells were pelleted and resuspended in Fixation buffer (420801, BioLegend) and incubated for 15' at 4 °C, washed twice with Intracellular Staining Permeabilisation Wash Buffer (421002, BioLegend) and stained overnight with the anti-dsRNA antibody J2 (English & Scientific Consulting) in Intracellular Staining Permeabilisation Wash Buffer. After washing, cells were incubated with anti-mouse Alexa Fluor 647 (ThermoFisher) for 1 h at room temperature and washed three times with Intracellular Staining Permeabilisation Wash Buffer before finally resuspending the cells in FACS buffer. Cells were analysed using a MACS Quant analyser 10 (Miltenyi), and data were processed using FlowJo software (Treestar). For immunofluorescence, cells were seeded on gelatine coated culture slides (Corning Falcon), washed twice with PBS with 2% FBS, fixed in fixation buffer (420801, BioLegend) and incubated for 15' at 4 °C. The fixed cells were washed three times with Intracellular Staining Permeabilisation Wash Buffer (421002, BioLegend) and stained overnight with the anti-dsRNA antibody J2. Cells were washed three time with Intracellular Staining Permeabilisation Wash Buffer and stained with anti-mouse FITC (406001, BioLegend) and incubated at RT for 1 h. After washing three times with Intracellular Staining Permeabilisation Wash Buffer, cells were stained with DAPI in antifade mounting media (Vectashield, Vector laboratories) and visualised using a fluorescence microscope (Axio, Carl Zeiss).

## Stable cell lines

Plasmids containing the sequences of mouse MDA5 (GE-healthcare, MMM1013-202762875) and 3xFLAG (pcDNA3.1-3xFLAG) were used to amplify Gibson assembly fragments to construct the MDA5 sequence with a 3xFLAG tag at the N-terminal end. A plasmid containing the MAVS sequence (GE-healthcare, MMM1013-202764911) was used as template to directly amplify the MAVS open reading frame with specific restriction sites and a N-terminal FLAG tag (for oligonucleotides, see Supplementary Data 5). The amplified and digested fragments were ligated into the pLenti-GIII-EF1a plasmid for constitutive expression and/or purification, and the pCW57-MCS1-P2A-MCS2 (89180, Addgene) plasmid, in which we replaced the hPGK promoter with the EF1a promoter for optimal expression in mESCs. For the ΔCTD mutant of MDA5, a reverse primer containing a stop codon and specific restriction site was used to amplify a fragment that was ligated into the same pLenti-GIII-EF1a and pCW57-MCS1-P2A-MCS2 plasmids. Verified

plasmids containing the genes of interest were transfected in mESCs using Lipofectamine 2000 and selected with the appropriate antibiotic. Clonal cell lines were isolated, expanded and tested for expression by qRT-PCR and Western blot. ShRNA sequences targeting IRF3, IRF7 and MAVS were designed using the Broad institute design tool (https://portals.broadinstitute.org/gpp/public/gene/search) and ligated into the pLKO.1 plasmid (24150, Addgene). After sequencing to verify integrity, the plasmids were transfected into mESCs using Lipofectamine 2000 and selected with the appropriate antibiotic. Clonal lines were isolated, expanded and tested for knock down of the target gene by RT-qPCR (Supplementary Data 5).

## MDA5 purification and poly(I:C) pulldown

HEK293T cells were transfected with the pLenti-GIII-EF1a plasmids expressing FLAG-MDA5 and FLAG-ΔCTD using Lipofectamine 2000, following manufacturer's instructions. Cells were collected after 48 h and resuspended in lysis buffer (20 mM HEPES-KOH pH 7.9, 100 mM KCl, 0.2 mM EDTA, 0.5 mM DTT, 0.2 mM PMSF, 5% glycerol, supplemented with Complete Protease inhibitors), followed by sonication. Lysates were incubated with pre-washed FLAG Dynabeads (Sigma-Aldrich, M8823) overnight. Beads were washed five times with lysis buffer, supplemented with 1 M KCl, with an additional final wash with TBS (50 mM Tris pH 7.4, 150 mM NaCl). Purified MDA5 proteins were eluted using FLAG peptide (Sigma-Aldrich, F4799) to a 100 μg/ml final concentration in TBS buffer, following manufacturer's instructions. Eluted proteins were quantified by Coomassie staining and nanodrop. For poly(I:C) pulldowns, poly(C)-agarose beads (Sigma-Aldrich, P9827) were washed twice with 50 mM Tris pH 7.0, 200 mM NaCl, and resuspended in 50 mM Tris-pH 7.0, 50 mM NaCl. Poly(I) (Sigma-Aldrich, P4154) was dissolved in 50 mM Tris-pH 7.0, 150 mM NaCl to a concentration of 2 mg/ml. One part of poly(C) beads and two parts of poly(I) solution were mixed and rocked for 1 h at 4 °C to form double-stranded RNA (dsRNA). Next, beads were pelleted, washed and resuspended in 50 mM Tris-pH 7.0, 150 mM NaCl. Poly(C) beads only were used as a control (single-stranded RNA, ssRNA). For pulldowns, equilibrated dsRNA-beads were resuspended in binding buffer (50 mM Tris-pH 7.5, 150 mM NaCl, 1 mM EDTA, 1% NP-40). A volume of beads slurry was combined with an equal volume of MDA5 protein (300 μg of WT or ΔCTD), plus protease/phosphatase inhibitors and 25 U RNAsin/ml. Reactions were incubated at 4 °C with gentle rotation for 1 h, washed three times with binding buffer and resuspended in SDS-PAGE (sodium dodecyl sulfate- polyacrylamide gel electrophoresis) loading buffer for downstream analyses by western blot.

## Total RNA high-throughput sequencing and analysis

Total RNA was extracted from cells using Tri-reagent and the quality assessed on the Agilent 2100 Electrophoresis Bioanalyser Instrument (Agilent, G2939AA). RNA was quantified using the Qubit 2.0 Fluorometer (ThermoFisher, Q32866). DNA contamination was quantified using the Qubit dsDNA HS assay kit and confirmed to be <6% (ThermoFisher, Q32854). Libraries for total RNA sequencing were prepared

using the NEBNext Ultra 2 Directional RNA library prep kit (Illumina, E7760) and the NEBNext rRNA Depletion kit (Human/Mouse/Rat) (Illumina, E6310) according to the provided protocol. Sequencing was performed on the NextSeq 2000 platform (Illumina, 20038897) using NextSeq 2000 P3 Reagents (200 Cycles, 2x100bp) (20040559). Raw sequences from the two sequencing runs were combined prior to being quality assessed using FASTQC (v0.11.8), and html format outputs generated. Cutadapt (v3.4) was used to trim the sequences with the following command: cutadapt -a AGATCGGAAGAG -A AGATCG-GAAGAG -j 0 -m 50 -o tr.$fq1 -p tr.$fq2 $fq1 $fq2 > tr.$sample.log. Sequences were not collapsed within each sample, and duplicates were retained. The murine genome (release 104, "REL104") and gtf formatted annotation was obtained by ftp from ensembl (ftp.ensembl.org/pub/release-104/fasta/mus_musculus). Alignment of reads (--very-sensitive -p 24 --no-mixed --no-discordant --no-unal -x $db −1 tr.$fq1 −2 tr.$fq2 2» $bt2log) to the REL104 reference set was performed using hisat2 (v 2.2.1). Alignments were stored in sorted, indexed BAM files. Raw tag counts per gene in each sample were extracted from the BAM files using the gtf file in combination with featureCounts function of Rsubread (featureCounts(files = bamfiles, annot.ext = 'Mus_musculus.104.gtf.gz', isGTFAnnotationFile = TRUE, countMultiMappingReads = TRUE, allowMultiOverlap = TRUE, isPairedEnd = TRUE, nthreads = 24, minOverlap = 10)), and were normalised as part of the DESeq2-based groupwise comparisons: no conversions were performed. Pairwise comparisons of sample groups were performed using the DESeq2 Bioconductor package, with alpha set at 1; as each sample group had 3 replicates, all groups were used. logFC values were shrunk using the ashr model in lfsShrink() and a significance threshold of $p < 0.01$ (adjusted) was applied. Gene ontology (biological process) analysis was performed using the ClueGo plugin in the Cytoscape package[86]. Cell type predictions were performed using the Mouse Gene Atlas within the Enrichr package (https://maayanlab.cloud/Enrichr/)[87]. Transcription factor predictions were done using the LOF dataset within the ShinyGO (0.81) package[49].

### Chromatin enrichment for proteomics (ChEP)

ChEP was conducted based on previously established protocols with slight modifications[54,88]. Briefly, $1 \times 10^7$ cells were crosslinked using 1% formaldehyde at room temperature (RT) for 10 min, followed by quenching with 0.25 M glycine. The cells were lysed in 1 ml ice-cold cell lysis buffer (25 mM Tris pH 7.4, 0.1% Triton X-100, 85 mM KCl, and 1× Roche protease inhibitor) and centrifuged at $2300 \times g$ for 5 min at 4 °C to isolate the nuclei. After discarding the cytoplasmic fraction, the pellet was resuspended in 500 µl SDS buffer (50 mM Tris pH 7.4, 10 mM EDTA, 4% SDS, and 1× Roche protease inhibitor) and incubated at RT for 10 min. The solution was diluted to 2 ml using urea buffer (10 mM Tris pH 7.4, 1 mM EDTA, and 8 M urea) and centrifuged at $16,100 \times g$ for 30 min at RT, repeating the step once. Subsequently, the pellet was resuspended with 2 ml SDS buffer and centrifuged at $16,100 \times g$ for 30 min at RT. The resulting protein pellet was dissolved in 250 µl storage buffer (10 mM Tris pH 7.4, 1 mM EDTA, 25 mM NaCl, 10% glycerol, and 1× Roche protease inhibitor) and sonicated using a Bioruptor (Diagenode) for five cycles (30 s on/off) to enhance solubility. Protein concentration was quantified via a Qubit protein assay (Invitrogen). For mass spectrometry (MS) analysis, 100 µg of protein extract was de-crosslinked at 95 °C for 30 min. Each timepoint included two biological replicates.

For proteomic analyses, de-crosslinked chromatin protein extracts (100 µg) were precipitated by adding 10 volumes of ice-cold 100% acetone and incubated at −20 °C for 15 min. The samples were centrifuged at $16,000 \times g$ for 10 min in a pre-cooled centrifuge. The resulting protein pellets were washed with 500 µl ice-cold acetone, centrifuged, and the acetone was discarded. The pellets were resuspended in 100 µl urea buffer (8 M urea, 100 mM Tris-HCl, pH 8.0, 10 mM DTT) and incubated at room temperature for 20 min. Subsequently, 10 µl iodoacetamide was added, and the mixture was

incubated in the dark for 15 min at room temperature. For protein purification and on-bead digestion[89], 5 µl of pre-washed carboxyl magnetic beads (Thermo Scientific) and 154 µl binding buffer (100% acetonitrile, ACN) were added, followed by incubation for 18 min at room temperature. The beads were washed twice with 180 µl of 70% ethanol and once with 200 µl of 100% ACN. Proteins were eluted and digested on beads in 100 µl of ammonium bicarbonate buffer (ABC) containing 3 µl trypsin (Promega) at 37 °C for 2 h. After removing the beads, the solution was incubated at 37 °C overnight to complete digestion. The digested peptides were fractionated using strong anion exchange into three fractions[90]: flow-through (FT), pH 8 elution, and pH 2 elution. The pH 8 and pH 2 fractions were combined. Finally, peptides were enriched and desalted using Stage-Tip[91] desalting before liquid chromatography-tandem mass spectrometry (LC-MS/MS) analysis using a Orbitrap Exploris 480 instrument.

Proteomic data analysis was carried out following previously described protocols with minor modifications[88]. Raw mass spectrometry data were processed using MaxQuant software[92] (v2.4.2) with searches performed against the mouse UniProt database (downloaded June 2017). Default parameters were used, with LFQ, iBAQ, and the "match between runs" feature enabled. Protein identification was validated against a decoy database generated within MaxQuant. Biological replicates were grouped to identify differentially expressed proteins. Data were filtered for three valid values in at least one group. Missing values were imputed using default settings in Perseus[93] (v2.0.11) based on the assumption that they were not detected because they were under or close to the detection limit. Differentially enriched proteins were determined using a two-tailed Student's $t$-test ($P < 0.05$) with a fold change threshold of >2. Differential proteins at 8, 24, and 48 h were excluded from the differential proteins at 0 h. Downstream analyses, including data visualisation, were performed using R software (version 4.4.1).

### Assay for Transposase-Accessible Chromatin using sequencing (ATAC-seq) and analysis

The ATAC (mitochondrial DNA-remove) protocol was adapted from previously established methods with modifications[94,95]. A total of 10,000 cells were collected, washed with 1 mL of ice-cold PBS containing EDTA-free protease inhibitor, and centrifuged at $500 \times g$ for 5 min at 4 °C. The supernatant was completely removed, and the cell pellet was resuspended in 20 µL of transposase reaction mixture, consisting of 1×Tagment DNA buffer, 1 µL Tn5 enzyme, and 0.02% digitonin. The transposition reaction was carried out at 37 °C for 30 min with continuous agitation. Following transposition, DNA was purified and the ATAC-seq library was constructed using the KAPA HyperPrep Kit (KAPA Biosystems, 07962363001) and barcoded with NEXTflex DNA barcodes (Integrated DNA Technologies). Paired-end sequencing of the library was performed on an Illumina NextSeq 500 platform.

Sequence reads were processed using the seq2science pipeline (v1.2.2)[96]. Paired-end reads were trimmed with fastp v0.23.2 with default options. Genome assembly GRCm39 was downloaded with genomepy 0.16.1. Reads were aligned with bwa-mem2 v2.2.1 with options '-M'. Afterwards, duplicate reads were marked with Picard MarkDuplicates v3.0.0. Before peak calling, paired-end info from reads was removed with seq2science so that both mates in a pair get used. Bam files were randomly sampled down to the reads number as the same as the lowest reads sample. Peaks were called with macs2 v2.2.7 with options ' --shift 75 --extsize 200 −nolambda'. Differential peaks were calculated by Diffbind (v3.12.0). Peak tracks and heatmaps were generated by Deeptools (v3.5.5). Peaks were annotated by ChIPseeker v1.38.0. GO analysis was conducted by clusterProfiler v4.10.1.

### Double-stranded RNA immunoprecipitations and analyses

Double stranded RNA was isolated using the dsRNA specific antibody J2 (English & Scientific Consulting). For each replicate, cells from one

confluent 10 cm plate were collected, washed twice with cold PBS and lysed in 1 ml IP-buffer (50 mM Tris pH 7.5, 150 mM NaCl, 1 mM EDTA, 1% Triton X-100) supplemented with protease inhibitors (S8820, Sigma-Aldrich). For each 1 ml of lysate, 10 μl DNAse (M6101, Promega) and 2 μl RNAse inhibitor (N2111, Promega) was added and incubated on ice for 30 min after which the lysate was spun at 13krpm, 4 °C for 15' and the supernatant collected. To couple J2 antibodies to magnetic protein G beads, 50 μl of protein G slurry (88848, Thermo Scientific) was washed three times with cold IP buffer, resuspend in 100 μl IP buffer with 5 μg of J2 antibody and mixed at 4 °C while rotating for 3 h. After incubation, the beads were washed three times with IP buffer, mixed with the lysate and incubated at 4 °C for 4 h while rotating. Beads were washed three times 5 min at 4 °C while rotating and tubes were changed once to prevent RNA carry over. After the last wash, the beads were resuspended in 250 μl of 0.3 M NaAc (pH 5.2) and RNA was extracted using Tri reagent LS according to the manufacturer's protocol. Purified RNA was sent for sequencing and processed according to the protocol described in the 'Total RNA high-throughput sequencing and analysis' section. Sequencing reads were processed, aligned and counted according to the protocol described in the 'Total RNA high-throughput sequencing and analysis' section. Normalisation (TMM method) and differential expression analysis between J2 IP and RNAseq gene counts were performed using the glmQLFit function from edgeR v. 4.2.2R package[97,98]. Transcripts were classified as enriched dsRNAs when J2 IP vs RNAseq log2FC > 0.75 and false discovery rate <0.05. The non-enriched transcripts were used as 'non-enriched control'. Sequences of processed transcripts (exons only) were obtained from genome reference GRCm39 using Ensembl genome annotation (release 104). Transcript lengths and lengths of 5' UTR, CDS and 3' UTR were also obtained from Ensembl release 104. Minimum free energy values were calculated via RNAfold from the ViennaRNA package v. 2.7.0 with default parameters[99]. Transposable elements (TE) annotation was obtained from the Hammel Lab (https://labshare.cshl.edu/shares/mhammelllab/www-data/TEtranscripts/TE_GTF/). Overlap between TE and exon coordinates was performed with findOverlaps function from the GenomicRanges v. 1.56.2 R package. Differences between RNA population features were calculated with one-tail Wilcoxon Test as implemented in wilcox_test function from rstatix v 0.7.2 package.

## 5'azacytidine treatment
WT mESCs (v6.5) were grown in the presence of different concentrations of 5'azacytidine (A2385, Sigma-Aldrich) for 48 h to allow for several replication cycles and depletion of DNA methylation. Cells were collected, RNA extracted and analysed as described above.

## Zebrafish experiments
Zebrafish wild-type strains AB/Tübingen/TAB (AB/Tu/TAB) were gifts from the zebrafish facilities at LARCEL (Málaga, Spain) and MPI-CBG (Dresden, Germany). The fish were maintained and bred under standard conditions[100]. Wild-type zebrafish embryos were obtained by natural mating of AB/Tu and TAB zebrafish of mixed ages (5–18 months). Pairs were randomly selected from 20 males and 20 females and zebrafish embryos were staged using the morphological criteria described before[101]. After fertilisation, eggs were collected and one-cell stage embryos were microinjected with 2 nL of 1 μg/μL poly(I:C) in PBS[71]. Control embryos were injected with the same volume of PBS. For Ruxolitinib experiments, embryos were first injected with poly(I:C) as described above and then incubated with 25 μM Ruxolitinib diluted in E3 medium 1-h post injection (as in ref.[102]). Control embryos were incubated in E3 containing an equivalent concentration of DMSO. Phenotypes were monitored at 24 h post fertilization (hpf).

For whole-mount in situ hybridisation, a partial region of Trim25 was cloned into pGEM-T using specific primers (Supplementary Data 5), linearised with NcoI and in vitro transcribed using SP6 (P1085, Promega) to generate an antisense trim25 probe, which was labelled with digoxigenin. Zebrafish embryos were fixed, hybridised, and stained using NBT/BCIP[103], imaged and analysed using ImageJ.

Total RNA was isolated from 50 to 60 embryos using Trizol (Invitrogen) according to the manufacturer's protocol. One microgram of RNA was DNase I treated and purified via phenol/chloroform extraction. cDNA was synthesised using the High-Capacity cDNA Reverse Transcription Kit (Applied Biosystems) and used for qPCR (GoTaq qPCR Mix, Promega).

Embryos of the same age were used to compare the differences in developmental defects due to their treatments. Embryos were classified based on severity of the development defects, where embryos were assigned class 1 defects if embryos exhibited failure to develop the head and class 2 defects if there were defects in trunk development.

## Reporting summary
Further information on research design is available in the Nature Portfolio Reporting Summary linked to this article.

## Data availability
The RNAseq data of MDA5 induction, dsRNA IP and ATACseq have been deposited on SRA under accession numbers PRJNA1219136, PRJNA1223341 and PRJNA1224926, respectively. The proteomics data has been deposited in the PRIDE database under accession number PXD059977. Existing datasets used in this study include the mouse ESC differentiation datasets E-MTAB-4904 and GSE127741. And zebrafish dataset GSE106430 [https://www.ncbi.nlm.nih.gov/bioproject/PRJNA416866]. Source data are provided with this paper.

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

## Acknowledgements

This work was funded by the Leverhulme Trust (RPG-2020-355) and the Wellcome Trust grants (221737/Z/20/Z) and (107665/Z/15/Z) to S.M. H.M. is supported by a Nederlandse Organisatie voor Wetenschappelijk Onderzoek (NWO) XL grant (OCENW.XL21.XL21.100) and a Radboud Science faculty grant (IRP voucher), Z.L. acknowledges support from the China Scholarship Council (CSC). C.R. is supported by EU funding under the MUR PNRR (Project no. E63C22001220001). T.T. is supported by AIRC under MFAG 2020 (ID. 24883 project) and by EU funding under the MUR PNRR 'National Center for Gene Therapy and Drugs based on RNA Technology' S6 RINGTAIL (Project no. CN00000041 CN3 RNA). Work in S.R.H group is supported by the Spanish Ministry of Science and Innovation (PID2020-115033RB-I00 and CNS2023-145402). We thank Ana Gazquez-Gutierrez for help with the J2-flow cytometry assay, Felix Mueller for help during revisions and Lisanne I. Knol for initial experiments with MDA5.

## Author contributions

S.M. and J.W. conceived the project and wrote the manuscript with the help from all authors. J.W. and J.L.W. performed most experiments with mESCs. A.I. performed computational analyses of RNA-sequencing datasets of mESCs. H.M., Z.L. and L.M. designed, performed and analysed the ATACseq and ChEP-MS experiments. T.T. and C.R. analysed the dsRNA immunoprecipitation datasets. A.A., P.G.M. and S.H. designed, performed and analysed the zebrafish experiments. E.T.F. performed analyses for *Ring1B* datasets.

## Competing interests

The authors declare no competing interests.
