## [Transparent Peer Review file · Nature Communications]

Double stranded RNA sensing is silenced during early embryonic development

Corresponding Author: Dr Sara Macias

Version 0:

Reviewer comments:

Reviewer #1

(Remarks to the Author)

The type I interferon (IFN) system is critical in vertebrate somatic cells to provide protection against viruses. One key intracellular innate receptor, MDA5, senses the presence within the cytosol of double-stranded RNA (dsRNA) of viral or endogenous source and triggers the IFN system. However, stem cells are endowed with an attenuated IFN response and therefore rely on alternative antiviral mechanisms for their protection. How and why the IFN system is attenuated in stem cells is not well understood.

In this study, the authors show that the silencing of MDA5 expression in mouse embryonic stem cells (mESCs) is crucial to prevent aberrant IFN response activation and this is essential to maintain pluripotency. First, the authors found that mESCs do not express crucial proteins from the RIG-I-like receptors (RLRs) pathway and found that MDA5 and IRF7 both become gradually expressed upon differentiation. They then use an inducible MDA5-expressing system to address the impact of MDA5 expression on mESCs. They show, by RNA-seq, that expression of MDA5 causes a development disruption with upregulation of genes involved in nervous system development while genes involved in maintaining mESCs features (e.g. pluripotency factors) were downregulated. They further find that these changes at the expression levels are accompanied by alterations of chromatin accessibility and chromatin-associated proteins in regions controlling nervous system functions, pluripotency and immune responses. The authors then show that mESCs express endogenous dsRNA in contrast to differentiated cells and that the expression of MDA5 induces the upregulation of several interferon-stimulated genes (ISGs). Further analyses show that the MDA5-mediated activation of the IRF3 pathway and not the IFN-mediated signalling pathway impact on the pluripotency programme of the cells. Finally, the importance of their findings was further shown by using a zebrafish model, in which the premature activation of the IFN response resulted in development defects.

This study is well-written and the findings are important and very interesting for the field of innate immunity and stem cell biology as it provides further key insights into the incompatibility between the IFN system and pluripotency. While this a comprehensive and thorough study, some findings need to be strengthened by further experimental evidence in order to fully support the conclusions of this study. The authors should therefore address the following comments:

Major comments:

Figure 4a: the authors nicely show that the induction of Flag-tagged MDA5 in mESCs results in the activation of a type I IFN response based on the upregulation of *ifnb* gene as well as three ISGs (*tnf*, *isg15* and *ifitm1*). While the upregulation of these 3 ISGs is compelling at the transcription level, it is unclear how robust is this response and whether the levels of ISG mRNAs observed upon MDA5 expression lead to readily detectable levels of the corresponding proteins as observed in viral infections or in autoinflammatory contexts. Given that the activation of the IFN response upon induction of MDA5 in mESCs constitutes a central point of the study, the authors should test whether the activation of the IFN response upon MDA5 expression leads to upregulation of ISGs at the protein level by western blot and/or ELISA/Multiplex assay kit (e.g. Meso Scale Discovery biomarker assay). As a general comment, this study monitors changes of gene expression by measuring mRNA levels, the authors should verify whether these changes occur at the protein level as well by analysing the level of

some ISGs-encoded proteins and pluripotency factors.

Figure 4c: the authors aimed to address whether the observed upregulation of ISGs upon the expression of MDA5 is caused by its well-established ability to sense endogenous dsRNA and consequently induce an IFN response. For this the authors engineered mESCs stable cell lines expressing a MDA5 mutant lacking its C-terminal domain known to be the region that recognises target RNAs. They showed that the induction of a truncated mutant of MDA5 did not lead to an upregulation of ISGs and also didn't cause changes in the levels of pluripotency factors (Nanog, Klf4, Pou5f1 in Figure 5b). Given that the induction of the full-length MDA5 protein was shown to induce an IFN response, the authors conclude that MDA5's ability to sense dsRNA is required to then initiate the signalling cascade and induce an IFN response. Yet, it is key to ensure that the expression level of the truncation mutant is comparable to the full-length to then compare their effects on the IFN pathway and the pluripotency state of the cells. To rule that the truncation mutant is simply too lowly expressed compared to full-length MDA5, the authors should provide a western blot comparing the expression levels of FLAG-MDA5 delta CTD versus FLAG-MDA5 WT at different time point post doxycycline treatment (0, 8, 24, 48 hrs post dox) and assess that the mESCs clones used express similar levels of FLAG-tagged MDA5. If it is indeed the case, the induction of *ifih1*, *ifnb1*, *isg15* and *ifitm1* as well as the pluripotency factors should be monitored by qRT-PCR and/or western blot within the same experiment to compare any changes at each time point caused by the mutant versus the full-length version of MDA5.

Figure 5a and b: To further address whether the observed upregulation of ISGs and the alteration of pluripotency upon MDA5 expression in mESCs is caused by the IFN induction pathway, the authors depleted essential components of the pathway using shRNA-mediated silencing in their inducible FLAG-MDA5 mESCs clone. They found that the upregulations of *ifnb1* and *tnf* as well as the downregulation of pluripotency factors were abolished upon silencing of IRF3 and MAVS, but not by knockdown of IRF7. These results suggest that the pathway leading to the production of IFN is necessary for the alterations observed in mESCs upon MDA5 expression. However, the authors showed in figure 5a that the induction of *ifih1* (encoding MDA5) expression is clearly attenuated especially in mESCs expressing shRNA targeting IRF3 where the induction upon doxycycline treatment appears to be of 2-fold versus 6 fold in cells expressing shRNA control (figure 5a, top left panel with *ifih1* level of expression assessed by qRT-PCR). Can the authors rule out that the loss of ISGs upregulation and pluripotency factors downregulation are not caused by a diminished induction of MDA5 in mESCs expressing shRNA against either IRF3 or MAVS? To circumvent this confounding effect, the authors could select a mESCs clone expressing shIRF3 and efficiently knocking down IRF3 but in which the level of MDA5 is induced at a similar level than in cells expressing the shRNA control.

Figure 5d: the authors nicely show that activation of the IFN signalling by treatment of mESCs with recombinant IFN- β doesn't lead to pluripotency dysregulation. Given all the changes the authors observed at the chromatin level upon MDA5 induction, the authors should strengthen their findings by analysing the importance of the IFN signalling in the same cellular context analysed throughout their study, e.g. in mESCs in which MDA5 expression is induced. For this, the authors inhibit the type I IFN receptor IFNAR with, for instance, the inhibitor Ruxolitinib and test whether the observed changes in ISGs as well as in the pluripotency factors do still occur upon MDA5 induction.

Minor comments:

- The current title "Double stranded RNA sensing drives interferon silencing in early development" is misleading. The study doesn't show that the sensing of endogenous dsRNA by MDA5 and the subsequent activation of the IFN response causes the silencing of the pathway. The authors should change their title to reflect more accurately their main findings.

- Line 78-80: "There is increasing evidence that transposable elements (TEs)...constitute a natural source of dsRNA in cells (6-8). Please mention the correct references here as the cited studies (Brownell et al., 2014; Xu et al. 2014; Smith et al., 2012) show the direct binding of IRF3 transcription factor to the promoter of some specific ISGs but do not look at the production of dsRNA from TEs.

- Line 158-161: The authors investigated the mechanism responsible for the silencing of *ifih1* gene in mESCs and tested the contribution of the Polycomb repressive complex 1 (PRC1). For this, the authors depleted one component of PRC1, RING1B, in a *Ring1A*^{-/-} ESCs. The authors should clarify in their manuscript the rationale behind using *Ring1A*^{-/-} cells to deplete RING1B and mention that both RING1A and RIG1B are functionally redundant Polycomb proteins and that the inactivation of the PRC1 requires the depletion of both proteins. Additionally, in extended data figure 1, the authors should show that RING1B depletion is indeed degran-depleted by, for instance, western blot to support their conclusion that PRC1 is not required for *ifih1* silencing.

- Please ensure that legends on the x-axis are present for each graphs (e.g legends on the x-axis in figures 1d, 4a, 4c are missing): and that the same labelling of x- and y-axis is used for similar analyses throughout the manuscript.

- Please ensure that the figures legends are clear and provide all the necessary information to understand the data shown. For example, the figure legends of Figure 5b and c doesn't specify the cells and the treatment (dox) used.

- Figure 1c: It is unclear how the statistical analyses show a significant difference in *ifnb1* and *cxcl10* gene expression upon treatment of mESCs with poly (I:C) while there is no upregulation shown in the graphs. The authors should perhaps consider using a log scale on the y-axis to show the differences occurring in mESC gene expression upon poly (I:C) treatment.

- Please display the size reference (molecular weight in kDalton) on all the western blot analyses shown in this study.
- Line 713, figure legend of Figure 2c: the authors mean “two lfh1-inducible” clones not genes, please clarify the figure legend.
- It is unclear what the extended figure 3c shows, please provide more information in the figure legend. In the figure legend, it is mentioned that the second clone doesn't harbour the same deletion. If it is an example of inaccessible region found after ATAC-seq analysis, why the authors chose a region that is not found as inaccessible in the second MDA5-expressing clone of this study? What do the authors mean by “second clone”, is it a the second MDA-5 expressing clone or a WT mESCs (v6.5 mESCs)? Please clarify accordingly in the corresponding legend. Shouldn't the control be an ATAC-seq analysis on MDA5 expressing clone before treatment with doxycycline?
- Please provide more clarification in the text and the figure legends of the data shown shown in figure 3h and 3i. In figure 3h, can the authors explain why they found an enrichment of DDX58 (encoding RIG-I) bound to chromatin upon MDA5 expression? To the knowledge of this reviewer, RIG-I is localised in the cytosol and is not associated to the chromatin. Is figure 3i showing the group of genes for which there is a positive correlation between early RNA-seq data and late ChEP-MS data? Many/most of the ISGs are not binding chromatin, why would an upregulation of innate immune genes correlate with these same proteins enriched with chromatin? Is the group of innate immune genes that are upregulated and whose encoded proteins is enriched to chromatin upon MDA5 induction (genes that are part of the "innate immune system" and cytokine signalling in immune system" in figure 3i) mainly consisting of transcription factors/repressors known to be involved in the control of immune genes? Please provide more details about the genes/proteins shown in figure 3i.
- Figure 4b: a faint band of MDA5 is detectable in the ssRNA IP, while a much robust band is observed upon dsRNA IP, which is consistent with the known ability of MDA5 to bind dsRNA. Yet, it is currently unclear if the blot shown in figure 4b is a single membrane or if there are two blots, one for the ssRNA pulldowns and the other for the dsRNA pulldowns. Can the authors clarify please? To assess that MDA5 preferentially bind dsRNA and that this binding is indeed lost when the C-terminal domain of MDA5 is lacking, the pulldowns of ssRNA and dsRNA with the two versions of MDA5 should be loaded on the same blot to then have the same exposure time and allow proper comparison.
- Figure 4f. The authors used as “Non-enriched control” the whole transcriptome obtained by RNA-seq. Can the authors explain why they didn't use an IgG control for this experiment to remove from their analysis any non-specific binding of RNA to the beads and antibody? Could the labelling be changed to “whole transcriptome” vs “anti-dsRNA-IP”? Finally, does “protein coding” RNAs include intronic regions?
- Please clarify the extended figure 6a: WT and MDA5-expressing clones were differentiated into embryoid bodies for 24 hours. Was the doxycycline treatment to induce MDA5 also performed for 24 hours and therefore were both the differentiation induction and doxycycline treatments induced at the same time? While the expression of MDA5 leads to the upregulation of differentiation markers (Sox2 and Hand1), can the authors explain why it also leads to upregulation of pluripotency markers (Nanog and Klf4) while they showed earlier in their study that MDA5 expression results in a decrease of pluripotency factors?
- The concentration of doxycycline used in the study is not mentioned in material and methods or in figure legends.
- Line 115: typo, change “supress” by “suppress”.

Reviewer #2

(Remarks to the Author)

Reviewer #3

(Remarks to the Author)

The manuscript by Macias et al. aims to dissect the mechanism in which the disable on type I interferon production in mESCs is due to the lack of dsRNA sensor MDA5. Ectopically expression of MDA5 sensitizes the mESC for interferon signaling activation which results in the damage of their stemness. Moreover, the high level of endogenous dsRNAs in mESC, which is normally absent in somatic cells, are responsible for triggering MDA5-mediated type I interferon pathway activation, leading to the upregulation of both interferon and ISGs. It has been shown that cells at early developmental stage including ESC and iPSC display some uniqueness on innate immune responses and modulation, including the resistance on viral infection with intrinsic ISG upregulation and others. The complexity of the cell destinies along differentiation and their different responses to IFN leave many open questions in this field. This manuscript adds some information for understanding the complicated scenario further, but the data is a little bit preliminary which hardly support a strong piece of theory add up to the existing understanding.

Here are a few of my comments in specific:

1. Figure 2, especially Figure 2c-d claimed that MDA5 overexpression induced a “neuronal-like” gene expression profile of ESC. As the treatment for cell clone screening easily affect the differential potential of ESCs, this experiment lacks necessary control (such as a MDA5 mutant or truncation) to prove that the expression profile change is indeed driven by MDA5, not a non-specific consequence of ectopic expression of proteins per se.
2. In Figure 3, the author showed that expression of MDA5 in ESCs results in innate immune responses, mainly rely on GO analysis from sequencing data. qPCR for signature genes of activated pathways, and the biochemistry approach for detecting the protein level of key players are also required to support the conclusion.
3. In Figure 4m, the author claimed that enriched dsRNAs harbour significantly more TEs per gene. As TEs with inverted sequences are capable to form dsRNA duplex, to further check which type of TEs enriched in ESCs specifically is suggested.
4. In Figure 5d, exogenous type I IFN stimulation led to upregulation of multiple ISGs such as Oas1, Stat1 and Isg15, and the author showed that sensing of exogenous IFN was unrelated to pluripotency perturbation. Since MDA5 also acting downstream of IFN signaling as an ISG, it will be worth to check if its expression could be induced by IFN treatment, the upregulation of which should impact the pluripotency according to the ectopic expression result.

Reviewer #4

(Remarks to the Author)

The manuscript by Witteveldt et al reports that the sensing of dsRNA by MDA5 during embryonic development is specifically silenced in order to allow for necessary gene expression patterns for proper development and differentiation. Specifically, the authors found that MDA5 is silenced early in embryogenesis, and that expression of MDA5 at this early stage leads to dysregulation of key gene expression patterns, driven by IRF3. Authors claim that MDA5 specifically requires silencing because ESCs inherently have high dsRNA that activates MDA5. They extend their findings to a zebrafish model, where they show similar inhibition of dsRNA sensing early in development.

This manuscript presents an interesting new discovery that MDA5 is specifically inhibited early in embryogenesis, and proposes that gene transcription driven by IRF3, rather than simply a toxic effect of interferon-driven gene expression, drives the deleterious effects. This novel finding is in theory quite attractive, but the manuscript has several points that need to be addressed to support the conclusions being made.

1. The biological model used for many of the experiments consists of an inducible MDA5 expression system. My concern with this model is that overexpression of MDA5 in many cells induces non-specific activation of the receptor. I imagine that the goal of this model is to express a biologically relevant amount of MDA5, that would be similar to the expression of MDA5 in the same cell type but at a later point of differentiation, e.g. ESCs at Day 6 of differentiation vs Day 0. Have the authors compared, either by WB or by RT-qPCR, the expression of MDA5 in their overexpressed early ESC to expression of later-stage differentiated cells, to see if their levels of MDA5 expression are comparable? A WB to show expression at protein levels would be particularly useful. Otherwise, it may be hard to draw conclusions about a natural state of ESCs being particularly prone to MDA5 activation due to, as they claim, high dsRNA levels, vs nonspecific effects from high MDA5 overexpression.
2. In Fig 2c, authors report on transcriptional changes resulting from MDA5 expression in ESCs. However, it is surprising that there were no obvious IFN/ISG expression patterns reported. One would expect that activation of MDA5 would lead to these sorts of transcriptional changes. Could author look at these specifically?
3. In Fig 3h, the authors do report an “innate immune signalling” pattern associated with MDA5 expression. Why was this not apparent in earlier datasets? Is this conclusion drawn from highly processed data points a valid analytical approach?
4. Authors state that in Extended Data 4 “ISG induction was also confirmed on the RNAseq data obtained from the two independent MDA5 overexpressing clones”, but this is not apparent. Can authors better indicate which of these genes are known ISGs? Furthermore, are the genes upregulated by PCR in Fig 4a not significantly differentially expressed in the RNAseq data? If so, why not?
5. I am unsure about the claim the authors make that a delta-CTD MDA5 results in no dsRNA binding. In fact, one of the papers cited by the authors (Peisley et al 2011 PNAS) specifically reports that the CTD is involved in cooperative filament assembly (i.e. MDA5 oligomerisation for activation) and does NOT affect dsRNA binding capacity. Therefore, use of a delta-CTD MDA5 is not really differentiating from an MDA5 that cannot bind to endogenous dsRNA, but rather one that has loss-of-function in downstream signalling. The data presented in their PolyIC pulldown is not well explained, and does not go far enough as to convince differently than what was previously reported. Rather, use of an MDA5 mutant that does not have dsRNA-binding capacity would be more appropriate. These have been reported by others:
<https://pmc.ncbi.nlm.nih.gov/articles/PMC5502429/>
<https://pubmed.ncbi.nlm.nih.gov/30449722/>
6. The authors claim that ESC have inherently high dsRNA loads, which is the driver of MDA5 activation and thus why these cells specifically require silencing of MDA5. However, the J2 flow cytometry on which this point strongly relies requires better controls. The comparison of the ESC is done to an unrelated cell type (a microglia). As the authors state in their

manuscript, there is high variability in dsRNA levels between cell types. Since the authors claim that ESCs in particular have high dsRNA, and that this is concordant with low MDA5, and that as ESCs differentiate this MDA5 expression increases, then is there a concordant drop in dsRNA load in these more differentiated cells? Performing the J2 staining on control cells that are closer to the ESC would be more a convincing comparison. Furthermore, the authors could consider doing J2 confocal microscopy as an additional experiment.

7. In Fig 4i, could the authors further analyse the data to report on e.g. introns vs exons, or 3/5'UTRs? TEs are often found in these regions, and increased pulldown of e.g. intron-containing mRNA, or extended 3'UTR would support their theory.

8. The data reported in Fig 4g-m is quite interesting, but requires better description of the analysis methods.

9. For Fig 5a, WBs are required to demonstrate knockdown of the target proteins.

10. The data in Fig 6 is a critical part of the author's proposal that activation of IRF3 is the driver for the gene dysregulation effects in ESCs. The treatment with IFN β is one way to test their theory, however, is it possible that a 4 hr treatment with IFN β may not be long enough to show an effect on the tested pluripotency genes? It is not clear in the legends how this compares to the length of dox treatment and the timing of MDA5 upregulation/activation. A more definitive approach could be to treat these MDA5-expressing cells with IFNAR-blocking antibodies to allow the activation of IRF3 and expression of genes, but block the IFN activity. Alternatively, authors could use IFNAR2-KO cells.

11. The zebrafish model presented in Fig 7 is an excellent development by the authors, but it could be further explored in order to strengthen their claims. For example, authors could include treatments similar to what was done for cells in Fig 6, e.g. exogenous IFN β treatment compared to dsRNA stimulation, and inclusion of an IFNAR2-KO or of IFNAR-blocking antibodies. The experiments as they stand cannot really disentangle the previously well-established damaging effects of interferon during early development vs the effects resulting from activation of MDA5.

Version 1:

Reviewer comments:

Reviewer #1

(Remarks to the Author)

All our comments have been thoroughly addressed with compelling experiments and additional supportive data, which have been incorporated into the manuscript.

Minor comments:

- -Figure 4b: Please display error bars as shown in other plots (e.g. figure 4a). Please also mention in the figure legend that the grey area depicts the threshold limit of the ELISA.
- -Figures 4d, 4e: we thank the authors to confirm that the MDA5 full-length version (FLAG-MDA5 WT) and the C-terminal deletion mutant of MDA5 (FLAG-MDA5 delta CTD) are expressed at similar levels by qRT-PCR (figure 4E). In figure 4D, the authors also confirm that the levels of protein of full-length MDA5 versus mutant at various times points are similar. These comparisons between protein levels can only be done if all samples are run in the same blots. In Supplementary figure 8.1d, we can see in the uncropped western blot that all the signals shown in Figure 4D are indeed derived from the same membrane and therefore the authors provided compelling evidence that the FLAG-MDA5 delta CTD and FLAG-MDA5 WT are indeed expressed at similar levels at both the RNA and protein level. The authors should specify in the figure legend that the two blots depicted in Figure 4D are signals from the same blot with the uncropped version shown in Supplementary Figure 8.1.
- Please ensure for consistency that throughout the manuscript the axes' labels are the same if the same analyses were performed. In Figure 7c, the y-axis is labelled as "expression levels (relative to gapdh)", while in Figure 1g the y-axis is labelled with "normalised to gapdh" only.
- Figure 6g: please specify which cells are used in the figure legend.
- Figure 4d,6b,6e: the molecular weight references (in kDalton) are still missing on the western blots.
- Figure 6f: perhaps add "MAVS ESC" instead of "MAVS" in the legend present within the figure to be consistent with the labels present in figure 6d.
- Figure 6h: the "w/o IFN" from the legend within the figure can be misleading as there is no IFN added in this experiment. This reviewer understands that the authors meant that there is no IFN signalling due to the presence of ruxolitinib but to avoid confusion please remove "w/o IFN" and leave "MDA (+dox +ruxo)" or perhaps add "MDA5 w/o IFN signalling (+dox +ruxo)".

Reviewer #2

(Remarks to the Author)

Reviewer #3

(Remarks to the Author)

The manuscript by Witteveldt et al. reported that the sensing of dsRNA by MDA5 is silenced during embryogenesis. With quite a lot of new evidence being added in the revised version, the updated manuscript has fulfilled my requests regarding to the previous reviewing points.

Reviewer #4

(Remarks to the Author)

I have carefully read the authors' responses to all reviewer comments. I find that the authors have addressed all of my concerns, and I have no further comments to the manuscript in its revised state.

Point-by-point response NCOMMS-25-23089-T

Reviewer #1 (Remarks to the Author):

The type I interferon (IFN) system is critical in vertebrate somatic cells to provide protection against viruses. One key intracellular innate receptor, MDA5, senses the presence within the cytosol of double-stranded RNA (dsRNA) of viral or endogenous source and triggers the IFN system. However, stem cells are endowed with an attenuated IFN response and therefore rely on alternative antiviral mechanisms for their protection. How and why the IFN system is attenuated in stem cells is not well understood.

In this study, the authors show that the silencing of MDA5 expression in mouse embryonic stem cells (mESCs) is crucial to prevent aberrant IFN response activation and this is essential to maintain pluripotency. First, the authors found that mESCs do not express crucial proteins from the RIG-I-like receptors (RLRs) pathway and found that MDA5 and IRF7 both become gradually expressed upon differentiation. They then use an inducible MDA5-expressing system to address the impact of MDA5 expression on mESCs. They show, by RNA-seq, that expression of MDA5 causes a development disruption with upregulation of genes involved in nervous system development while genes involved in maintaining mESCs features (e.g. pluripotency factors) were downregulated. They further find that these changes at the expression levels are accompanied by alterations of chromatin accessibility and chromatin-associated proteins in regions controlling nervous system functions, pluripotency and immune responses. The authors then show that mESCs express endogenous dsRNA in contrast to differentiated cells and that the expression of MDA5 induces the upregulation of several interferon-stimulated genes (ISGs). Further analyses show that the MDA5-mediated activation of the IRF3 pathway and not the IFN-mediated signalling pathway impact on the pluripotency programme of the cells. Finally, the importance of their findings was further shown by using a zebrafish model, in which the premature activation of the IFN response resulted in development defects.

This study is well-written and the findings are important and very interesting for the field of innate immunity and stem cell biology as it provides further key insights into the incompatibility between the IFN system and pluripotency. While this a comprehensive and thorough study, some findings need to be strengthened by further experimental evidence in order to fully support the conclusions of this study. The authors should therefore address the following comments:

We thank the reviewer for their constructive comments. The suggested experiments have strengthened the conclusions of our study.

Major comments:

1. Figure 4a: the authors nicely show that the induction of Flag-tagged MDA5 in mESCs results in the activation of a type I IFN response based on the upregulation of ifnb gene as well as three ISGs (tnf, isg15 and ifitm1). While the upregulation of these 3 ISGs is compelling at the transcription level, it is unclear how robust is this response and whether the levels of ISG mRNAs observed upon MDA5 expression lead to readily detectable levels of the corresponding proteins as observed in viral infections or in autoinflammatory contexts. Given that the activation of the IFN response upon induction of MDA5 in mESCs constitutes a central point of the study, the authors should test whether the activation of the IFN response upon MDA5 expression leads to upregulation of ISGs at the protein level by

western blot and/or ELISA/Multiplex assay kit (e.g. Meso Scale Discovery biomarker assay). As a general comment, this study monitors changes of gene expression by measuring mRNA levels, the authors should verify whether these changes occur at the protein level as well by analysing the level of some ISGs-encoded proteins and pluripotency factors.

We have now tested IFN- β protein production by ELISA with both inducible and constitutively expressed MDA5 in ESCs (new Fig 4B). Inducible MDA5 results in IFN- β protein production, slightly over the detection limit (~ 8.5-9 pg/ml). Constitutively expressed MDA5 results in a more pronounced production of IFN- β (~13 pg/ml). Although the constitutively expressed form leads to a more obvious immune phenotype, these cells cannot be indefinitely propagated, as they eventually silence MDA5 expression after few passages. We have also compared constitutively expressed MDA5 with the Δ CTD mutant. In agreement with our previous results, overexpression of the Δ CTD mutant of MDA5 does not result in IFN- β protein production (new Fig 4B, lower panel).

We have also tried to assess the expression of some ISGs by western blot, including RIG-I and OASL. RIG-I increase was detected by mass spectrometry analyses (Fig 3i). However, due to problems with the sensitivity of the antibodies, the generally low expression of ISGs, the short-lived nature of MDA5 expression and the attenuated response of ESCs to IFNs (see Fig 1h), these were, unfortunately, undetectable by western blot.

We have also measured the protein levels of key pluripotency markers, including Nanog and KLF4, both of which showed marked decreases in expression during induction of MDA5 (new Fig 3e).

2. Figure 4c: the authors aimed to address whether the observed upregulation of ISGs upon the expression of MDA5 is caused by its well-established ability to sense endogenous dsRNA and consequently induce an IFN response. For this the authors engineered mESCs stable cell lines expressing a MDA5 mutant lacking its C-terminal domain known to be the region that recognises target RNAs. They showed that the induction of a truncated mutant of MDA5 did not lead to an upregulation of ISGs and also didn't cause changes in the levels of pluripotency factors (Nanog, Klf4, Pou5f1 in Figure 5b). Given that the induction of the full-length MDA5 protein was shown to induce an IFN response, the authors conclude that MDA5's ability to sense dsRNA is required to then initiate the signalling cascade and induce an IFN response. Yet, it is key to ensure that the expression level of the truncation mutant is comparable to the full-length to then compare their effects on the IFN pathway and the pluripotency state of the cells. To rule that the truncation mutant is simply too lowly expressed compared to full-length MDA5, the authors should provide a western blot comparing the expression levels of FLAG-MDA5 delta CTD versus FLAG-MDA5 WT at different time point post doxycycline treatment (0, 8, 24, 48 hrs post dox) and assess that the mESCs clones used express similar levels of FLAG-tagged MDA5. If it is indeed the case, the induction of ifih1, ifnb1, isg15 and ifitm1 as well as the pluripotency factors should be monitored by qRT-PCR and/or western blot within the same experiment to compare any changes at each time point caused by the mutant versus the full-length version of MDA5.

We have now compared the expression levels of WT and Δ CTD MDA5, (new Figure 4d and 4e, respectively) and observed comparable levels of expression on both protein

and transcript levels. Using the same samples, we have further confirmed that only WT MDA5 is capable of inducing IFN and ISG expression (new Figure 4e), but not the Δ CTD mutant. In agreement, only WT MDA5 perturbs the expression of pluripotency factors and differentiation markers (new Figure 6d and Suppl. Figure 2b)

*3. Figure 5a and b: To further address whether the observed upregulation of ISGs and the alteration of pluripotency upon MDA5 expression in mESCs is caused by the IFN induction pathway, the authors depleted essential components of the pathway using shRNA-mediated silencing in their inducible FLAG-MDA5 mESCs clone. They found that the upregulations of *ifnb1* and *tnf* as well as the downregulation of pluripotency factors were abolished upon silencing of IRF3 and MAVS, but not by knockdown of IRF7. These results suggest that the pathway leading to the production of IFN is necessary for the alterations observed in mESCs upon MDA5 expression. However, the authors showed in figure 5a that the induction of *ifih1* (encoding MDA5) expression is clearly attenuated especially in mESCs expressing shRNA targeting IRF3 where the induction upon doxycycline treatment appears to be of 2-fold versus 6 fold in cells expressing shRNA control (figure 5a, top left panel with *Ifih1* level of expression assessed by qRT-PCR). Can the authors rule out that the loss of ISGs upregulation and pluripotency factors downregulation are not caused by a diminished induction of MDA5 in mESCs expressing shRNA against either IRF3 or MAVS? To circumvent this confounding effect, the authors could select a mESCs clone expressing shIRF3 and efficiently knocking down IRF3 but in which the level of MDA5 is induced at a similar level than in cells expressing the shRNA control.*

We have now compared the induction of FLAG-MDA5 in all shRNA lines by western blot and confirm that there are no apparent differences in protein expression levels (new Fig. 6b). In addition, we are including new western blot analyses to confirm successful depletion of IRF3, IR7 and MAVS (new Fig. 6a).

4. Figure 5d: the authors nicely show that activation of the IFN signalling by treatment of mESCs with recombinant IFN- β doesn't lead to pluripotency dysregulation. Given all the changes the authors observed at the chromatin level upon MDA5 induction, the authors should strengthen their findings by analysing the importance of the IFN signalling in the same cellular context analysed throughout their study, e.g. in mESCs in which MDA5 expression is induced. For this, the authors inhibit the type I IFN receptor IFNAR with, for instance, the inhibitor Ruxolitinib and test whether the observed changes in ISGs as well as in the pluripotency factors do still occur upon MDA5 induction.

As suggested by the reviewer, we have blocked IFN signalling using Ruxolitinib. Despite blocking IFN signalling, the expression of pluripotency genes is still significantly down regulated upon MDA5 induction (new Fig. 6h). We have also performed a similar experiment in zebrafish (new Fig 7f-g). Both experiments suggest that IFN signalling is not required to observe the developmental phenotypes.

Minor comments:

- The current title "Double stranded RNA sensing drives interferon silencing in early development" is misleading. The study doesn't show that the sensing of endogenous dsRNA by MDA5 and the subsequent activation of the IFN response causes the silencing of the pathway. The authors should change their title to reflect more accurately their main findings.

We agree with the reviewer and we have therefore changed the title to: ‘Silencing of double stranded RNA sensing is required for early embryonic development’

- Line 78-80: “There is increasing evidence that transposable elements (TEs)...constitute a natural source of dsRNA in cells (6-8). Please mention the correct references here as the cited studies (Brownell et al., 2014; Xu et al. 2014); Smith et al., 2012) show the direct binding of IRF3 transcription factor to the promoter of some specific ISGs but do not look at the production of dsRNA from TEs.

These references have now been corrected to: Chiappinelli (Cell 2015; <https://pmc.ncbi.nlm.nih.gov/articles/PMC4556003/>), Roulois (Cell 2015; <https://pmc.ncbi.nlm.nih.gov/articles/PMC4556003/>), Mehdipour (Nature 2020; <https://pmc.ncbi.nlm.nih.gov/articles/PMC4556003/>).

- Line 158-161: The authors investigated the mechanism responsible for the silencing of Ifih1 gene in mESCs and tested the contribution of the Polycomb repressive complex 1 (PRC1). For this, the authors depleted one component of PRC1, RING1B, in a Ring1A -/- ESCs. The authors should clarify in their manuscript the rationale behind using Ring1A-/- cells to deplete RING1B and mention that both RING1A and RIG1B are functionally redundant Polycomb proteins and that the inactivation of the PRC1 requires the depletion of both proteins. Additionally, in extended data figure 1, the authors should show that RINGB1 depletion is indeed degra-deleted by, for instance, western blot to support their conclusion that PRC1 is not required for Ifih1 silencing.

The PRC1 complex is a E3 ubiquitin-ligase that monoUb H2A at K119. Although depleting the catalytic subunit of the complex RING1B is enough to observe depletion of ubH2A, the double knock-out of RING1B and its homolog RING1A is required to totally erase this mark

(<https://www.sciencedirect.com/science/article/pii/S1534580704003673?via%3Dihub>).

The functional redundancy of these two genes has now been highlighted in the text.

The RING1B-degron dataset is published and it has been extensively used. For more details see: (<https://www.nature.com/articles/s41556-024-01493-w> ; <https://www.nature.com/articles/s41594-021-00661-y> ; <https://www.sciencedirect.com/science/article/pii/S2211124719317140?via%3Dihub>)

- Please ensure that legends on the x-axis are present for each graphs (e.g legends on the x-axis in figures 1d, 4a, 4c are missing): and that the same labelling of x- and y-axis is used for similar analyses throughout the manuscript.

Corrected. Please note that the legends in 1d, 4a and 4c were not missing. These data are not represented as fold-change (FC) but normalised to *Gapdh* or *Actb*.

- Please ensure that the figures legends are clear and provide all the necessary information to understand the data shown. For example, the figure legends of Figure 5b and c doesn't specify the cells and the treatment (dox) used.

These have now been corrected. The concentration of doxycycline used is 250ng/ml. This information has been added to material and methods.

*- Figure 1c: It is unclear how the statistical analyses show a significant difference in *ifnb1* and *cxcl10* gene expression upon treatment of mESCs with poly (I:C) while there is no*

upregulation shown in the graphs. The authors should perhaps consider using a log scale on the y-axis to show the differences occurring in mESC gene expression upon poly (I:C) treatment.

Corrected. These data have now been represented using a log scale (new Fig 1G).

- Please display the size reference (molecular weight in kDalton) on all the western blot analyses shown in this study.

Molecular weight sizes have now been included in all blots.

- Line 713, figure legend of Figure 2c: the authors mean "two *lfih1*-inducible" clones not genes, please clarify the figure legend.

This has now been corrected

- It is unclear what the extended figure 3c shows, please provide more information in the figure legend. In the figure legend, it is mentioned that the second clone doesn't harbour the same deletion. If it is an example of inaccessible region found after ATAC-seq analysis, why the authors chose a region that is not found as inaccessible in the second MDA5-expressing clone of this study? What do the authors mean by "second clone", is it a the second MDA-5 expressing clone or a WT mESCs (v6.5 mESCs)? Please clarify accordingly in the corresponding legend. Shouldn't the control be an ATAC-seq analysis on MDA5 expressing clone before treatment with doxycycline?

Apologies for the confusion. After performing ATAC-seq analyses, we realised that the MDA5 clone (1) has a small genomic deletion, probably caused by insertion of the doxycycline-inducible form of MDA5. Also, our analysis suggests that the MDA5 clone (2), which was also used for the time course RNA-seq analyses, does not contain this deletion. All these together suggest that the defects observed in the developmental programme upon MDA5 induction are not indirectly caused by the microdeletion, since both clones behave similarly. These observations have now been included in the figure legend.

With regards the ATAC control, we have analysed the ATAC-seq data using both normalisation to WT (v6.5) ESCs and non-dox MDA5 ESCs. We noticed some leaky expression of MDA5 in the absence of doxycycline, making the changes induced upon MDA5 expression less obvious.

- Please provide more clarification in the text and the figure legends of the data shown shown in figure 3h and 3i. In figure 3h, can the authors explain why they found an enrichment of *DDX58* (encoding RIG-I) bound to chromatin upon MDA5 expression? To the knowledge of this reviewer, RIG-I is localised in the cytosol and is not associated to the chromatin. Is figure 3i showing the group of genes for which there is a positive correlation between early RNA-seq data and late ChEP-MS data? Many/most of the ISGs are not binding chromatin, why would an upregulation of innate immune genes correlate with these same proteins enriched with chromatin? Is the group of innate immune genes that are upregulated and whose encoded proteins is enriched to chromatin upon MDA5 induction (genes that are part of the "innate immune system" and cytokine signalling in immune system" in figure 3i) mainly consisting of transcription factors/repressors known to be involved in the control of immune genes? Please provide more details about the genes/proteins shown in figure 3i.

The ChEP-MS protocol performs an enrichment for chromatin-bound proteins to facilitate their detection. However, this protocol will still detect abundant non-chromatin bound proteins. In support of this, the manuscript that originally developed this protocol detected 3,522 proteins after chromatin enrichment, which agrees with this protocol being an enrichment step rather than being a pure 'only chromatin-bound' fraction (<https://pmc.ncbi.nlm.nih.gov/articles/PMC4300392/>). This protocol has also been used to compare chromatin-enriched factors in naïve vs primed hPSCs. A total of 4,576 proteins were detected after enrichment (<https://www.nature.com/articles/s41556-022-00932-w>). We hypothesise that the induction of RIG-I is enough to detect it after chromatin-enrichment, but it is not necessarily directly associated with chromatin. The same rationale would apply for other regulators of the IFN response that are not known to associate with chromatin, such as Vimentin (see Fig 3i-j). However, it is important to note that RIG-I has been shown to be able to localise in the nucleus in some settings (PMID: 35598407).

Innate immune genes that correlate at the RNA and protein level (Fig 3i-j) are Hck, Myo1c, Dusp6, Sugt1, Anxa2, Ddx58, Pnp2 and Pnp (annotated in Reactome, the R-MMU-168249 Innate Immune system). Using GO Biological pathways, we obtained similar genes: Hck, Ddx58, Epha2, Myo1c, Vim, Foxp1, Trim35, Anxa3, Anxa2, Ass1, Cdkn1c and Pnp2.

To confirm that the innate immune activation is also observed in the RNA-seq datasets, we have included the infection/immune pathways obtained by KEGG analyses (Suppl Fig 3e).

- Figure 4b: a faint band of MDA5 is detectable in the ssRNA IP, while a much robust band is observed upon dsRNA IP, which is consistent with the known ability of MDA5 to bind dsRNA. Yet, it is currently unclear if the blot shown in figure 4b is a single membrane or if there are two blots, one for the ssRNA pulldowns and the other for the dsRNA pulldowns. Can the authors clarify please? To assess that MDA5 preferentially bind dsRNA and that this binding is indeed lost when the C-terminal domain of MDA5 is lacking, the pulldowns of ssRNA and dsRNA with the two versions of MDA5 should be loaded on the same blot to then have the same exposure time and allow proper comparison.

This is a single membrane. The uncropped blot has now been included in Suppl. Fig 8. Apologies for forgetting to include the uncropped version in the initial submission.

- Figure 4f. The authors used as "Non-enriched control" the whole transcriptome obtained by RNA-seq. Can the authors explain why they didn't use an IgG control for this experiment to remove from their analysis any non-specific binding of RNA to the beads and antibody? Could the labelling be changed to "whole transcriptome" vs "anti-dsRNA-IP"? Finally, does "protein coding" RNAs include intronic regions?

Our initial experiments included IgG as a negative control, however no RNA was recovered and library preparation failed. Therefore, we decided to use the input to calculate the enrichment for each transcript in the dsRNA IP. This approach has been successfully used in the past by Tebaldi's team for dsRNA identification (<https://www.sciencedirect.com/science/article/pii/S2666166721000733> and <https://pubmed.ncbi.nlm.nih.gov/32497523/>). The enrichment over input approach identifies two groups of transcripts; first, those enriched in the dsRNA IP, and

second, those not enriched. These are the two subgroups that have been consistently used in Figure 5 (non-enriched control vs enriched dsRNA).

Since we have used transcripts, these did not contain introns. However, we agree with the reviewer that this is an important distinction we should address and have now reanalysed the datasets and quantified the number of intronic reads in input vs IP. We have identified that the dsRNA IPs contain, proportionally, more reads mapping to introns than input, suggesting that introns are also prone to generate dsRNA.

- Please clarify the extended figure 6a: WT and MDA5-expressing clones were differentiated into embryoid bodies for 24 hours. Was the doxycycline treatment to induce MDA5 also performed for 24 hours and therefore were both the differentiation induction and doxycycline treatments induced at the same time? While the expression of MDA5 leads to the upregulation of differentiation markers (Sox2 and Hand1), can the authors explain why it also leads to upregulation of pluripotency markers (Nanog and Klf4) while they showed earlier in their study that MDA5 expression results in a decrease of pluripotency factors?

ESCs, with or without MDA5, were differentiated for 24 in the presence of doxycycline, we have clarified this in the legend (Suppl. Fig 7). There are two different defects caused by MDA5 induction. First, MDA5 in ESCs causes the downregulation in the steady-state levels of pluripotency factors (Figure 3D-E). However, when ESCs are forced to differentiate (Suppl. Fig. 7), MDA5 impairs the normal silencing of the pluripotency factors, suggesting that it also interferes with differentiation. Therefore, we do not expect to observe a downregulation in the pluripotency factors in any of the conditions of the Suppl. Fig. 7 upon MDA5 induction, as none represents normally grown ESCs with MDA5.

- The concentration of doxycycline used in the study is not mentioned in material and methods or in figure legends.

This has now been included in material and methods.

- Line 115: typo, change “supress” by “suppress”.

Corrected

Reviewer #2 (Remarks to the Author):

Reviewer #3 (Remarks to the Author):

The manuscript by Macias et al. aims to dissect the mechanism in which the disable on type I interferon production in mESCs is due to the lack of dsRNA sensor MDA5. Ectopically expression of MDA5 sensitizes the mESC for interferon signaling activation which results in the damage of their stemness. Moreover, the high level of endogenous dsRNAs in mESC, which is normally absent in somatic cells, are responsible for triggering MDA5-mediated type I interferon pathway activation, leading to the upregulation of both interferon and ISGs. It has

been shown that cells at early developmental stage including ESC and iPSC display some uniqueness on innate immune responses and modulation, including the resistance on viral infection with intrinsic ISG upregulation and others. The complexity of the cell destinies along differentiation and their different responses to IFN leave many open questions in this field. This manuscript adds some information for understanding the complicated scenario further, but the data is a little bit preliminary which hardly support a strong piece of theory add up to the existing understanding.

Here are a few of my comments in specific:

1. Figure 2, especially Figure 2c-d claimed that MDA5 overexpression induced a “neuronal-like” gene expression profile of ESC. As the treatment for cell clone screening easily affect the differential potential of ESCs, this experiment lacks necessary control (such as a MDA5 mutant or truncation) to prove that the expression profile change is indeed driven by MDA5, not a non-specific consequence of ectopic expression of proteins per se.

We have now included the log₂FC expression of several neuronal markers in both MDA5 clones using RNA-seq (new Figure 2e). We have also confirmed the induction of these neuronal markers by RT-qPCR, and confirmed that only WT MDA5, but not the Δ CTD mutant, can induce their expression (new Suppl. Fig 2b).

2. In Figure 3, the author showed that expression of MDA5 in ESCs results in innate immune responses, mainly rely on GO analysis from sequencing data. qPCR for signature genes of activated pathways, and the biochemistry approach for detecting the protein level of key players are also required to support the conclusion.

We have now included a new panel of ISGs induced by MDA5 in both individual clones by RNA-seq (new Suppl. Fig 4a). We have also confirmed their induction both at the transcript and protein level. For transcripts, we have performed RT-qPCR for a panel of ISGs during a time course of MDA5 induction (new Fig 4e). As a control, we have also confirmed that the Δ CTD-MDA5 mutant fails to induce these ISGs, as expected (new Fig 4e). At the protein level, we have confirmed that both inducible and constitutively expressed MDA5 in ESCs results in IFN- β production by ELISA. Please see also response to reviewer 1, point 1.

3. In Figure 4m, the author claimed that enriched dsRNAs harbour significantly more TEs per gene. As TEs with inverted sequences are capable to form dsRNA duplex, to further check which type of TEs enriched in ESCs specifically is suggested.

We have now performed a more detailed analysis and confirmed that retrotransposons are enriched in the dsRNA IP. These belong to the SINEs and LTRs class (new Fig 5m). More specifically, these belonged to the B1 (Alu), B2 and B4 families as well as ERVL-MaLR (new Suppl. Fig 5). Importantly, B1 (Alu) and B2 are the most enriched and also known to be inverted repeats.

4. In Figure 5d, exogenous type I IFN stimulation led to upregulation of multiple ISGs such as *Oas1*, *Stat1* and *Isg15*, and the author showed that sensing of exogenous IFN was unrelated to pluripotency perturbation. Since MDA5 also acting downstream of IFN signaling as an ISG, it will be worth to check if its expression could be induced by IFN treatment, the upregulation of which should impact the pluripotency according to the ectopic expression result.

We have now confirmed that exogenous IFN cannot induce endogenous MDA5 expression (new Fig 1e). We conclude that MDA5 is silenced in ESCs and IFN stimulation is not enough to derepress its silencing.

Reviewer #4 (Remarks to the Author):

The manuscript by Witteveldt et al reports that the sensing of dsRNA by MDA5 during embryonic development is specifically silenced in order to allow for necessary gene expression patterns for proper development and differentiation. Specifically, the authors found that MDA5 is silenced early in embryogenesis, and that expression of MDA5 at this early stage leads to dysregulation of key gene expression patterns, driven by IRF3. Authors claim that MDA5 specifically requires silencing because ESCs inherently have high dsRNA that activates MDA5. They extend their findings to a zebrafish model, where they show similar inhibition of dsRNA sensing early in development.

This manuscript presents an interesting new discovery that MDA5 is specifically inhibited early in embryogenesis, and proposes that gene transcription driven by IRF3, rather than simply a toxic effect of interferon-driven gene expression, drives the deleterious effects. This novel finding is in theory quite attractive, but the manuscript has several points that need to be addressed to support the conclusions being made.

1. The biological model used for many of the experiments consists of an inducible MDA5 expression system. My concern with this model is that overexpression of MDA5 in many cells induces non-specific activation of the receptor. I imagine that the goal of this model is to express a biologically relevant amount of MDA5, that would be similar to the expression of MDA5 in the same cell type but at a later point of differentiation, e.g. ESCs at Day 6 of differentiation vs Day 0. Have the authors compared, either by WB or by RT-qPCR, the expression of MDA5 in their overexpressed early ESC to expression of later-stage differentiated cells, to see if their levels of MDA5 expression are comparable? A WB to show expression at protein levels would be particularly useful. Otherwise, it may be hard to draw conclusions about a natural state of ESCs being particularly prone to MDA5 activation due to, as they claim, high dsRNA levels, vs nonspecific effects from high MDA5 overexpression.

All our results lead us to hypothesise that overexpression of MDA5 induces the IFN response due to dsRNA recognition. First, overexpression of an MDA5 mutant that fails to bind dsRNA and oligomerise does not result in IFN activation, despite expression levels being similar to WT MDA5 (new Fig, 4d-e). Second, overexpression of the adaptor protein of MDA5, MAVS, that also aggregates its CARD domains for signalling transduction, does not result in IFN activation.

We agree with the reviewer that overexpression of MDA5 in other cell lines can also lead to signalling. However, we believe this signalling is specific, and caused by varying degrees of accumulation of dsRNAs in different cell types. Most established cell lines in laboratories are derived from tumours. Tumours tend to have alterations in epigenetic silencing, and increased expression of transposable elements, which could possibly lead to dsRNA formation (<https://pubmed.ncbi.nlm.nih.gov/33888553/>). For instance, HeLa or PA-1 cells have detectable levels of dsRNA as shown by immunofluorescence or flow cytometry (<https://pubmed.ncbi.nlm.nih.gov/30046113/> and <https://www.biorxiv.org/content/10.1101/2025.05.28.656609v1.full>).

We have now compared the absolute expression levels of MDA5 in *in vitro* differentiated ESCs vs the induced MDA5 clones (1) and (2) after 8 hours of doxycycline by RT-qPCR. These results lead us to conclude that in our system there are no aberrantly high levels of overexpression of MDA5.

MDA5 (*Ifih1*) transcript levels were compared after normalisation to *Gapdh*. Samples analysed are normal ESCs (ESCs) and *in vitro* differentiated ESCs (diff), compared to clones (1) and (2) of MDA5 expressing cells after 8 hours of doxycycline addition.

2. In Fig 2c, authors report on transcriptional changes resulting from MDA5 expression in ESCs. However, it is surprising that there were no obvious IFN/ISG expression patterns reported. One would expect that activation of MDA5 would lead to these sorts of transcriptional changes. Could author look at these specifically? 3. In Fig 3h, the authors do report an “innate immune signalling” pattern associated with MDA5 expression. Why was this not apparent in earlier datasets? Is this conclusion drawn from highly processed data points a valid analytical approach? 4. Authors state that in Extended Data 4 “ISG induction was also confirmed on the RNAseq data obtained from the two independent MDA5 overexpressing clones”, but this is not apparent. Can authors better indicate which of these genes are known ISGs? Furthermore, are the genes upregulated by PCR in Fig 4a not significantly differentially expressed in the RNAseq data? If so, why not

Apologies if this part was not reported clearly. To resolve any confusion, we now have included graphs to represent the enrichment for infection/immune pathways obtained with the RNA-seq timecourse dataset (new Suppl Fig 3e). We have also included a new panel of ISGs induced by MDA5 in both individual clones by RNA-seq (new Suppl Fig 3f).

At the protein level, we have confirmed that both inducible and constitutively expressed MDA5 in ESCs results in IFN- β production by ELISA (new Fig 4b). Please see also response to reviewer 1, point 1 and reviewer 2, point 2.

In addition, we have explored the potential for the changes in gene expression induced by MDA5 to be caused by type I IFN expression using the Interferome database <https://interferome.org/interferome/home.jsp>. From the 396 genes differentially expressed at t=4h, 269 are predicted to be regulated by type-I IFNs. From the 368 genes commonly dysregulated at t=8, 245 are predicted to be regulated by type I IFNs. Similarly, from the 248 genes dysregulated at t=24h, 152 are regulated by type I IFN.

To end, in our experience detecting the expression of IFN- β and ISGs by RNA-seq in cell lines that are not IFN-proficient is challenging due to their generally low expression, which can lead to variability and non-significant enrichment. This is why we favour quantifying these genes by alternative more sensitive and reliable methods, such as RT-qPCR.

5. I am unsure about the claim the authors make that a delta-CTD MDA5 results in no dsRNA binding. In fact, one of the papers cited by the authors (Peisley et al 2011 PNAS)

specifically reports that the CTD is involved in cooperative filament assembly (i.e. MDA5 oligomerisation for activation) and does NOT affect dsRNA binding capacity. Therefore, use of a delta-CTD MDA5 is not really differentiating from an MDA5 that cannot bind to endogenous dsRNA, but rather one that has loss-of-function in downstream signalling. The data presented in their PolyIC pulldown is not well explained, and does not go far enough as to convince differently than what was previously reported. Rather, use of an MDA5 mutant that does not have dsRNA-binding capacity would be more appropriate. These have been reported by others:

<https://pmc.ncbi.nlm.nih.gov/articles/PMC5502429/>

<https://pubmed.ncbi.nlm.nih.gov/30449722/>

We would like to thank the reviewer for pointing out the complexity of the mode of binding of MDA5 to dsRNA. Both the paper mentioned by the reviewer, which uses cryo-EM of MDA5-dsRNA filaments (<https://pubmed.ncbi.nlm.nih.gov/30449722/>) as well as another paper published on the same year using a crystallography approach (<https://pubmed.ncbi.nlm.nih.gov/23273991/>) show that all domains of MDA5 (Hel1, Hel2i, Hel2 and CTD) interact with dsRNA. MDA5 forms a ring around the dsRNA molecule, and different regions from all domains are also required to oligomerise. Therefore, the ability to bind dsRNA and to oligomerise are intimately connected, and to our knowledge there are no known mutants that can provide a separation of function. For instance, the second paper mentioned by the reviewer (<https://pubmed.ncbi.nlm.nih.gov/28606988/>) describes a loss-of-function mutation in MDA5. The mutation is a single amino acid substitution in Hel1 (K365E), and the mutation is predicted to abolish dsRNA binding and oligomerisation. Quoting from the paper '*Substitution with glutamic acid introduces a negative charge that is predicted to abolish the protein–nucleic acid interaction and MDA5 oligomerization (Fig. 2 C)... suggesting that the defect of this mutant lies in MDA5 dimerization or oligomerization upon binding or detecting RNA, although we did not test this further.* Using the same polyIC pulldown approach we used, this manuscript demonstrates that the point mutation abolished stable binding or oligomerisation to the dsRNA polyIC, thus resulting in defective interferon signalling.

Similarly, the CTD seems to be required for both dsRNA binding as well as oligomerisation. Although in isolation the CTD from MDA5 binds dsRNA less efficiently than the CTD from RIG-I and LGP2 (Figure 1C, <https://pmc.ncbi.nlm.nih.gov/articles/PMC2719387/>), removing the CTD from MDA5 decreases its binding affinity to dsRNA (Figure 1B and FigureS1B with helicase+CTD and Figure S1D with helicase only, <https://pubmed.ncbi.nlm.nih.gov/22314235/>). These results agree with most domains from MDA5 being required to initially bind dsRNA. Second, despite the challenges of generating mutants that uncouple dsRNA binding from oligomerisation, it is important to note that oligomerisation can only happen on a dsRNA substrate, and not in the absence of RNA (<https://pubmed.ncbi.nlm.nih.gov/30449722/>, Figure S7C), suggesting that MDA5 can only trigger the IFN response upon dsRNA binding and oligomerisation (or filament formation). Finally, further confirming the critical role for oligomerisation in the discrimination of bona-fide MDA5 substrates, MDA5 has been shown to be capable of binding dsRNA, dsDNA and ssRNA *in vitro*, however, only binding to dsRNA is cooperative (multimeric), and capable of downstream signalling (Fig 1, <https://pubmed.ncbi.nlm.nih.gov/22160685/>).

For all these reasons, we believe that overexpression of the MDA5- Δ CTD is a useful control to test if the negative impact of MDA5 on pluripotency requires dsRNA binding and oligomerisation. This control demonstrates that the defects observed are not an indirect consequence of overexpressing MDA5 in ESCs, but that this is mediated by dsRNA recognition, oligomerisation and signalling. We will be modifying the text to make this concept clearer.

The polyIC pulldown assay is used to test the ability of proteins to bind dsRNA (polyIC) vs ssRNA (polyC). This assay has been successfully used in the past to test MDA5 dsRNA binding ability (<https://pubmed.ncbi.nlm.nih.gov/16116171/> or <https://pubmed.ncbi.nlm.nih.gov/28606988/>). This approach cannot discriminate if MDA5 is binding as a monomer or is oligomerising on the dsRNA. Considering the current literature, we predict that upon polyIC binding WT-MDA5 will be oligomerising, thus resulting in a strong pulldown with polyIC, while the Δ CTD, which it is known to bind with less affinity to dsRNA and fails to oligomerise, will result in decreased pulled-down levels with polyIC, as we observe.

6. The authors claim that ESC have inherently high dsRNA loads, which is the driver of MDA5 activation and thus why these cells specifically require silencing of MDA5. However, the J2 flow cytometry on which this point strongly relies requires better controls. The comparison of the ESC is done to an unrelated cell type (a microglia). As the authors state in their manuscript, there is high variability in dsRNA levels between cell types. Since the authors claim that ESCs in particular have high dsRNA, and that this is concordant with low MDA5, and that as ESCs differentiate this MDA5 expression increases, then is there a concordant drop in dsRNA load in these more differentiated cells? Performing the J2 staining on control cells that are closer to the ESC would be more a convincing comparison. Furthermore, the authors could consider doing J2 confocal microscopy as an additional experiment.

To address this point we have performed immunofluorescence with anti-dsRNA J2 antibody in ESC vs *in vitro*-differentiated ESCs. Quantification of dsRNA positive cells confirmed the results from the Flow Cytometry experiment that ESCs accumulates more endogenous dsRNA compared to IFN-competent cells (new Fig 5c-e).

7. In Fig 4i, could the authors further analyse the data to report on e.g. introns vs exons, or 3/5'UTRs? TEs are often found in these regions, and increased pulldown of e.g. intron-containing mRNA, or extended 3'UTR would support their theory.

We have attempted to perform these analyses and we do not observed differences in enrichment between the different mRNA regions (5'UTR, CDS, 3'UTR). However, we had anticipated that this analysis would not be very informative. DsRNA IPs pull-down the whole transcript, which can contain both dsRNA and ssRNA regions. J2-based IPs do not provide positional information of which region is actually forming the dsRNA within the transcript. However, we have now calculated the density of intronic sequences and confirmed that dsRNA IPs are more enriched in intronic regions than input, confirming the suggestion of this reviewer.

8. The data reported in Fig 4g-m is quite interesting, but requires better description of the analysis methods.

More details have been added, please see ‘double-stranded RNA immunoprecipitation and analyses’ section in material and methods.

9. For Fig 5a, WBs are required to demonstrate knockdown of the target proteins.

We have now included WBs confirming the successful depletion of IRF3, IRF7 and MAVS (new Fig. 6a). We have also confirmed that all these depleted cell lines express similar levels of FLAG-MDA5 (new Fig. 6b).

10. The data in Fig 6 is a critical part of the author’s proposal that activation of IRF3 is the driver for the gene dysregulation effects in ESCs. The treatment with IFN β is one way to test their theory, however, is it possible that a 4 hr treatment with IFN β may not be long enough to show an effect on the tested pluripotency genes? It is not clear in the legends how this compares to the length of dox treatment and the timing of MDA5 upregulation/activation. A more definitive approach could be to treat these MDA5-expressing cells with IFNAR-blocking antibodies to allow the activation of IRF3 and expression of genes, but block the IFN activity. Alternatively, authors could use IFNAR2-KO cells.

We have now tested this possibility using ruxolitinib, which is an inhibitor of IFN signalling. Despite blocking IFN signalling and ISG production, the expression of MDA5 still results in significant downregulation of the expression of pluripotency genes (new Fig. 6h). A similar approach has been used for zebrafish, see below.

11. The zebrafish model presented in Fig 7 is an excellent development by the authors, but it could be further explored in order to strengthen their claims. For example, authors could include treatments similar to what was done for cells in Fig 6, e.g. exogenous IFN β treatment compared to dsRNA stimulation, and inclusion of an IFNAR2-KO or of IFNAR-blocking antibodies. The experiments as they stand cannot really disentangle the previously well-established damaging effects of interferon during early development vs the effects resulting from activation of MDA5.

We have tested if the dsRNA-induced phenotype is affected by blocking the IFN response with ruxolitinib. First of all, we tested if ruxolitinib is functional in zebrafish and used concentrations reported to drive differential susceptibility to viruses in zebrafish (25 μ M; <https://pubmed.ncbi.nlm.nih.gov/39861877/>). We observed a reduction in the expression of the ISG *Trim25* using ruxolitinib (new Figure 7f). This experiment suggest that blocking the IFN response using ruxolitinib does not rescue the dsRNA-induced phenotype (Fig 7g).

We also considered stimulating the fish with type I IFN, but unfortunately, zebrafish type I IFN is not commercially available to perform this type of assays.

Point-by-point response to the reviewers' comments

Reviewer #1 (Remarks to the Author):

All our comments have been thoroughly addressed with compelling experiments and additional supportive data, which have been incorporated into the manuscript.

Minor comments:

- -Figure 4b: Please display error bars as shown in other plots (e.g.figure 4a). Please also mention in the figure legend that the grey area depicts the threshold limit of the ELISA.

Error bars have been added and legend has been modified.

- -Figures 4d, 4e: we thank the authors to confirm that the MDA5 full-length version (FLAG-MDA5 WT)and the C-terminal deletion mutant of MDA5 (FLAG-MDA5 delta CTD) are expressed at similar levels by qRT-PCR (figure 4E). In figure 4D, the authors also confirm that the levels of protein of full-length MDA5 versus mutant at various times points are similar. These comparisons between protein levels can only be done if all samples are run in the same blots. In Supplementary figure 8.1d, we can see in the uncropped western blot that all the signals shown in Figure 4D are indeed derived from the same membrane and therefore the authors provided compelling evidence that the FLAG-MDA5 delta CTD and FLAG-MDA5 WT are indeed expressed at similar levels at both the RNA and protein level. The authors should specify in the figure legend that the two blots depicted in Figure 4D are signals from the same blot with the uncropped version shown in Supplementary Figure 8.1.

Legend of figure 4d has been modified to reflect the reviewers suggestion.

- Please ensure for consistency that throughout the manuscript the axes' labels are the same if the same analyses were performed. In Figure 7c, the y-axis is labelled as "expression levels (relative to gapdh)", while in Figure 1g the y-axis is labelled with "normalised to gapdh" only.

Axis labels have been checked and modified when necessary to ensure a consistent nomenclature.

- Figure 6g: please specify which cells are used in the figure legend.

The name of the cell line used in figure 6g has been added

- Figure 4d,6b,6e: the molecular weight references (in kDalton) are still missing on the western blots.

Size references have been added to the Western blots.

- Figure 6f: perhaps add " MAVS ESC" instead of "MAVS" in the legend present within the figure to be consistent with the labels present in figure 6d.

Naming in Figure 6f has been modified

- Figure 6h: the “w/o IFN” from the legend within the figure can be misleading as there is no IFN added in this experiment. This reviewer understands that the authors meant that there is no IFN signalling due to the presence of ruxolitinib but to avoid confusion please remove “w/o IFN” and leave “MDA (+dox +ruxo)” or perhaps add “MDA5 w/o IFN signalling (+dox +ruxo)”.

We have changed the name to the latter suggestion provided by the reviewer.

Reviewer #2 (Remarks to the Author):

Reviewer #3 (Remarks to the Author):

The manuscript by Witteveldt et al. reported that the sensing of dsRNA by MDA5 is silenced during embryogenesis. With quite a lot of new evidence being added in the revised version, the updated manuscript has fulfilled my requests regarding to the previous reviewing points.

Reviewer #4 (Remarks to the Author):

I have carefully read the authors' responses to all reviewer comments. I find that the authors have addressed all of my concerns, and I have no further comments to the manuscript in it's revised state.